# Transferable Adversarial Attacks on SAM and Its Downstream Models

**Song Xia[1], Wenhan Yang[2], Yi Yu[1], Xun Lin[3], Henghui Ding[4],**
**Lingyu Duan[2,5], Xudong Jiang[1]***

[1]Nanyang Technological University, [2]Pengcheng Laboratory,
[3]Beihang University, [4]Fudan University, [5]Peking University
{xias0002,yuyi0010,exdjiang}@ntu.edu.sg, yangwh@pcl.ac.cn,
linxun@buaa.edu.cn, hhding@fudan.edu.cn, lingyupku@edu.cn

## Abstract

The utilization of large foundational models has a dilemma: while fine-tuning downstream tasks from them holds promise for making use of the well-generalized knowledge in practical applications, their open accessibility also poses threats of adverse usage. This paper, for the first time, explores the feasibility of adversarial attacking various downstream models fine-tuned from the segment anything model (SAM), by solely utilizing the information from the open-sourced SAM. In contrast to prevailing transfer-based adversarial attacks, we demonstrate the existence of adversarial dangers even without accessing the downstream task and dataset to train a similar surrogate model. To enhance the effectiveness of the adversarial attack towards models fine-tuned on unknown datasets, we propose a universal meta-initialization (UMI) algorithm to extract the intrinsic vulnerability inherent in the foundation model, which is then utilized as the prior knowledge to guide the generation of adversarial perturbations. Moreover, by formulating the gradient difference in the attacking process between the open-sourced SAM and its fine-tuned downstream models, we theoretically demonstrate that a deviation occurs in the adversarial update direction by directly maximizing the distance of encoded feature embeddings in the open-sourced SAM. Consequently, we propose a gradient robust loss that simulates the associated uncertainty with gradient-based noise augmentation to enhance the robustness of generated adversarial examples (AEs) towards this deviation, thus improving the transferability. Extensive experiments demonstrate the effectiveness of the proposed universal meta-initialized and gradient robust adversarial attack (UMI-GRAT) toward SAMs and their downstream models. Code is available at https://github.com/xiasong0501/GRAT.

## 1 Introduction

Large foundation models that are trained on a broad scale of data have gained massive success in various applications [6], such as vision-language chatbot [1], text-image generation [42, 44, 46], image-grounded text generation [2, 31], and anything segmentation [26]. The segment anything model (SAM) [26], trained on vast amounts of data from the SA-1B dataset, is capable of handling diverse and complex visual tasks. The open accessibility of SAM makes it a promising foundation model, serving as the starting point for fine-tuning analytics models in certain domains and downstream applications, *e.g.,* medical segmentation [61, 54, 39], 3D object segmentation [9], camouflaged object segmentation [11], overhead image segmentation [45], and high-quality segmentation [25]. However, many studies [5, 18, 57, 38, 27, 55, 58, 66, 65, 59, 48] have highlighted the secure issues of deep learning models towards adversarial attacks. By corrupting the clean input with a finely crafted and

---

*Corresponding author (exdjiang@ntu.edu.sg)

38th Conference on Neural Information Processing Systems (NeurIPS 2024).

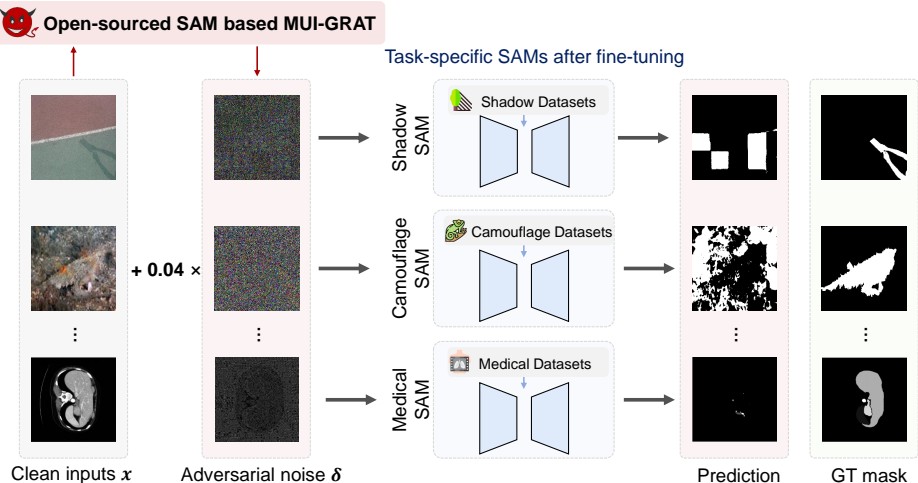

Figure 1: An illustration of UMI-GRAT towards SAM and its downstream tasks. The UMI-GRAT can mislead various downstream models by solely utilizing information from the open-sourced SAM.

nearly imperceptible adversarial perturbation, the attacker can mislead the well-trained model at a high success rate, with limited information available (*e.g.,* the surrogate model or limited queries). Consequently, significant concerns arise regarding fine-tuning open-sourced models, as it inevitably leaks critical information on downstream models, increasing their vulnerability to adversarial attacks.

Existing adversarial attacks can be roughly categorized into white-box attacks [3, 18] and black-box attacks [56, 23], based on whether the attacker can fully access the victim model. The pre-requisite of fully accessing the victim model complicates the practical deployment of white-box attacks. Conversely, transfer-based black-box attacks that require less information pose a substantial threat to real-world applications. The prevalent transfer-based black-box adversarial approaches [8, 12, 17, 22, 32, 34, 36, 15, 60] typically suppose strong prior knowledge of the victim model's task and training data, such as a 1000-class classification task in ImageNet, thereby facilitating the training of a similar surrogate model to generate potent adversarial examples (AEs). However, few studies [64, 63] consider a more practical and challenging scenario wherein the attacker is unaware of the victim model's tasks and the associated training data, due to stringent privacy and security protection policies (*e.g.,* datasets containing medical or human facial information). Moreover, the increasing size of large foundation models significantly amplifies the costs of training effective task-specific surrogate models. Thus, a more general and practical security concern is to explore the capability of attackers to mount adversarial attacks on any victim model even without the need of accessing to its downstream tasks-specific datasets to train a closely aligned surrogate model.

Given the practical security concerns of utilizing large foundation models, this paper investigates the potential risks associated with fine-tuning the open-sourced SAM on a private and encrypted dataset. We introduce a strong transfer-based adversarial attack called universal meta-initialized and gradient robust adversarial attack (UMI-GRAT), which effectively mislead SAMs and their various fine-tuned models without accessing the downstream task and training dataset, as illustrated in Figure 1.

The contributions of our paper are summarized as follows:

- We begin an investigation into a more practical while challenging adversarial attack problem: attacking various SAMs' downstream models by solely utilizing the information from the open-sourced SAMs. We provide the theoretical insights and build the experimental setting and benchmark, aiming to serve as a preliminary exploration for future research.
- We propose an offline universal meta-initialization (UMI) algorithm to extract the intrinsic vulnerability inherent in the foundation model, which is utilized as prior knowledge to enhance the effectiveness of adversarial attacks through meta-initialization.
- We theoretically formulate that, when using the open-sourced SAM as the surrogate model, a deviation occurs inevitably from the optimal direction of updating the adversary. Correspondingly, we propose a gradient robust loss to mitigate this deviation.
- Extensive experiments demonstrate the effectiveness of the proposed UMI-GRAT toward SAMs and its downstream models. Moreover, UMI-GRAT can serve as a plug-and-play strategy to significantly improve current state-of-the-art transfer-based adversarial attacks.

## 2 Related Work

The adversarial attacks aim to mislead the target (or victim) model by adding a small adversarial perturbation in the clean input. Existing black-box attacks can be broadly categorized into query-based and transfer-based adversarial attacks.

### 2.1 Query-based black box attacks

The query-based attacks consider the scenarios where the attacker does not have enough information to train a satisfying surrogate model, thus generating adversarial examples by interacting and analyzing the outputs from the victim model. This kind of attack can be divided into score-based query attacks (SQAs) [12, 24, 47] that update the AEs by observing the change of the model's prediction (*e.g.,* the logits or softmax probability) and decision-based query attacks (DQAs) [7, 10] that only rely on the model's top-1 prediction to update the AEs. However, this black-box search is naturally the NP-hard problem, and solving the optimal update strategy is non-differentiable. This makes query-based attack requires thousands of interactions with the victim model, making query-based attacks characterized by low throughput, high latency, and marked conspicuousness to attack real-world deployed systems.

### 2.2 Transfer-based black box attacks

The transfer-based adversarial attacks generate the AEs to mislead the victim model based on a similar surrogate model. Existing work mainly focuses on improving the transferability of AEs, which can be categorized into four groups: input-augmentation-based attacks [8, 56, 52] that enhances the effectiveness of generated AEs by augmenting the clean input (*e.g.,* using crop or rotation), optimization-based attacks [13, 35, 51, 62] that utilizes a better optimization strategy to guide the update of AEs, model modification-based attacks [53, 4] or ensemble-based attacks [19, 43, 37, 33] that enhances the AEs by utilizing a more powerful surrogate model, and feature-based attacks [32, 34] that attack the extracted feature in the intermediate layer. However, most of the work makes a strong assumption that the surrogate and victim models are optimized for the same task with identical data distribution, for example, both surrogate and victim models are optimized on the ImageNet dataset to complete the classification task. In real-world deployed systems, due to privacy and security concerns, attackers typically cannot access the training data (*e.g.,* datasets containing private information) or obtain the optimization objectives of the victim model, making training a similar surrogate model exceedingly difficult and unfeasible.

To address the challenge of deploying transfer-based black-box attacks without knowing the victim model's task and training dataset, this paper investigates the feasibility of attacking any victim model optimized for unknown tasks and distributions that significantly differ from the open-sourced surrogate model. We provide both theoretical and analytical evidence demonstrating that our proposed method can enhance the transferability and effectiveness of the generated AEs. Moreover, the proposed MUI-GRAT can serve as a plug-and-play adversarial generation strategy to enhance most existing transfer-based adversarial attacks for this challenging task.

## 3 Preliminaries

### 3.1 Adversarial attacks

Let $f$ be any deep learning model and $\mathcal{L}$ be the loss function (*e.g.,,* the cross-entropy loss) that evaluates the quality of the model's prediction. Let $\mathcal{B}_\epsilon(\boldsymbol{x}) = \left\{ \boldsymbol{x}' : \|\boldsymbol{x}' - \boldsymbol{x}\|_p \leq \epsilon \right\}$ be an $\ell_p$-norm ball centered at the input $\boldsymbol{x}$, where $\epsilon$ is a pre-defined perturbation bound. For each input $\boldsymbol{x}$, the untargeted adversarial attacks aim to find an adversarial perturbation $\boldsymbol{\delta}$ by solving:

$$\max_{\boldsymbol{x}+\boldsymbol{\delta}\in\mathcal{B}_\epsilon(\boldsymbol{x})} \mathcal{L}\left(f\left(\boldsymbol{x}\right), f\left(\boldsymbol{x}+\boldsymbol{\delta}\right)\right). \tag{1}$$

An effective solution to Equation 1 is iteratively updating the adversarial perturbation $\boldsymbol{\delta}$ based on the gradient of the loss function, for example, the iterative fast sign gradient method (I-FGSM) [28], which iteratively updates $\boldsymbol{\delta}$ by:

$$\boldsymbol{\delta}_{t+1} = clip_{\mathcal{B}_\epsilon}\{\boldsymbol{\delta}_t + \alpha \cdot sign\left(\nabla_{\boldsymbol{\delta}_t}\mathcal{L}\left(f\left(\boldsymbol{x}\right), f\left(\boldsymbol{x}+\boldsymbol{\delta}_t\right)\right)\right)\}, \tag{2}$$

where $\nabla$ calculates the gradient and $sign$ returns the sign (*i.e.*,-1 or +1). $\alpha$ is a pre-defined step size to update the adversarial perturbation. $clip$ constrains the magnitude of the perturbation by projecting $\boldsymbol{\delta}$ into the boundary of the $\ell_p$-norm ball $\mathcal{B}_\epsilon$.

## 3.2 Segment anything model

The SAM consists of three parts: an image encoder $f_{\phi_{im}}$, a lightweight prompt encoder $f_{\phi_{pt}}$, and a lightweight mask decoder $f_{\phi_{mk}}$. SAM gives the mask prediction based on the image input $\boldsymbol{x}$ and prompt input $\boldsymbol{p}$, which is expressed as:

$$\boldsymbol{y} = \mathcal{SAM}(\boldsymbol{x}, \boldsymbol{p}) = f_{\phi_{mk}} \left( f_{\phi_{im}}(\boldsymbol{x}), f_{\phi_{pt}}(\boldsymbol{p}) \right), \tag{3}$$

where $f_{\phi_{im}}$ is the image encoder that provides the fundamental understanding by converting natural images into feature embeddings and $f_{\phi_{pt}}$ extracts prompt embeddings. $f_{\phi_{mk}}$ is the mask decoder that gives the mask prediction by fusing the information from both feature and prompt embeddings.

# 4 Methodology

## 4.1 Problem formulation

Let $f_{\phi_s}$ denote the foundation model trained on a general dataset $D$, and $f_{\phi_\tau}$ denote the victim model fine-tuned on any downstream dataset $D_\tau$, the parameters of those two models typically satisfy that:

$$\phi_s = \arg\min_{\phi_s} \mathop{\mathbb{E}}_{(\boldsymbol{x}, \boldsymbol{y}) \sim D} \left[ \mathcal{L}\left( f_{\phi_s}(\boldsymbol{x}), \boldsymbol{y} \right) \right]; \phi_\tau = \arg\min_{\phi_\tau} \mathop{\mathbb{E}}_{(\boldsymbol{x}_\tau, \boldsymbol{y}_\tau) \sim D_\tau} \left[ \mathcal{L}_\tau \left( f_{\phi_\tau}(\boldsymbol{x}_\tau), \boldsymbol{y}_\tau \right) \right], \phi_\tau \stackrel{initia}{\longleftarrow} \phi_s \tag{4}$$

**Definition 1 (Transferable adversarial attack via open-sourced SAM).** *For any SAM's downstream model $f_{\phi_\tau}$ and the clean input $x_\tau$, without any further information on the downstream task and dataset, the attacker aims to find the adversarial perturbation $\boldsymbol{\delta}_s$ such that:*

$$\max_{\boldsymbol{\delta}_s \in \mathcal{B}_\epsilon} \mathcal{L}_\tau \left( f_{\phi_\tau}(\boldsymbol{x}_\tau), f_{\phi_\tau}(\boldsymbol{x}_\tau + \boldsymbol{\delta}_s) \right) \text{ s.t. } \{\boldsymbol{\delta}_s = \mathcal{AT}(f_{\phi_s}, \boldsymbol{x}_\tau), Private(D_\tau)\}, \tag{5}$$

*where $\mathcal{AT}$ is the adversarial attack strategy and $f_{\phi_s}$ is the open-sourced SAM. A solution to that is fine-tuning an optimal surrogate model $f_{\phi_s^*}$ that closely aligns with the victim model. However, this approach becomes extremely challenging, when the downstream dataset is inaccessible to the attacker. Alternatively, an effective solution is to design an optimal attack strategy $\mathcal{AT}^*$ such that:*

$$\mathcal{AT}^* = \arg\max_{\mathcal{AT}} \mathop{\mathbb{E}}_{(\boldsymbol{x}_\tau, \boldsymbol{y}_\tau) \sim D_\tau} \left[ \mathcal{L}_\tau \left( f_{\phi_\tau}(\boldsymbol{x}_\tau), f_{\phi_\tau}(\boldsymbol{x}_\tau + \mathcal{AT}^*(f_{\phi_s}, \boldsymbol{x}_\tau)) \right) \right]. \tag{6}$$

Notably, $f_{\phi_s}$ and $f_{\phi_\tau}$ are optimized on two distinctive distributions $D$ and $D_\tau$ with losses $\mathcal{L}$ and $\mathcal{L}_\tau$, leading to a significant input-output mapping gap and gradient disparity, such as $Cosine\_similarity(\nabla f_{\phi_s}(\boldsymbol{x}_\tau), \nabla f_{\phi_\tau}(\boldsymbol{x}_\tau)) \ll 1$. This misalignment critically undermines the effectiveness of current gradient-based adversarial attack strategies.

**Further analysis on attacking SAMs.** The standard operation to deploy SAM on downstream tasks $\tau$ involves fine-tuning the image encoder $f_{\phi_{im}}$ to inherit some well-generalized knowledge. Concurrently, the lightweight prompt encoder $f_{\phi_{pt}}$ and the mask decoder $f_{\phi_{mk}}$ are trained from scratch to better accommodate the task. Considering the pivotal importance of feature embeddings and the substantial variation caused by full retraining, an intuitive approach to generate effective adversarial perturbation $\boldsymbol{\delta}_s$ is utilizing the common information in $f_{\phi_{im}}$ to achieve:

$$\max_{\boldsymbol{\delta}_s \in \mathcal{B}_\epsilon} \mathcal{L} \left( f_{\phi_{im}^\tau}(\boldsymbol{x}_\tau), f_{\phi_{im}^\tau}(\boldsymbol{x}_\tau + \boldsymbol{\delta}_s) \right) \text{ s.t. } \boldsymbol{\delta}_s = \mathcal{AT}^*(f_{\phi_{im}}, \boldsymbol{x}_\tau), \tag{7}$$

where $\phi_{im}^\tau$ denotes the updated parameters for the downstream model's image encoder after fine-tuning. Unless otherwise specified, we denote $\phi_{im}$ as $\phi$ in our subsequent content and take the general SAM image encoder $f_\phi$ as the surrogate model $f_{\phi_s}$. We aim to generate the transfer-based AEs to attack any fine-tuned image encoder $f_{\phi_\tau}$ on task $\tau$, thereby misleading the entire prediction.

## 4.2 Extract the intrinsic vulnerability via universal meta initialization

Considering the great variation brought by fine-tuning the model on a new task $\tau$, we aim to extract the intrinsic vulnerability of the foundational model that remains invariant after fine-tuning. Subsequently, this extracted vulnerability is leveraged as prior knowledge to initialize and enhance the adversarial

attack $\mathcal{AT}^*$. Inspired by the universal adversarial perturbation [40] that maintains effectiveness across various inputs, we propose the universal meta initialization (UMI) algorithm, which optimizes the initialization of adversarial perturbation to ensure both effectiveness and fast adaptability by meta-learning [41, 16]. We define the universal and meta-initialized perturbation $\boldsymbol{\delta}$ as follows.

**Definition 2** (**Universal and meta-initialized perturbation $\boldsymbol{\delta}$**). *Given the foundation model $f_\phi$ and its fine-tuned models $f_{\phi_\tau}$ on downstream tasks $\tau$, the universal and meta-initialized perturbation $\boldsymbol{\delta}$ that extracts the intrinsic vulnerability ensures both effectiveness and fast adaptability, which are:*

1. ***Effectiveness (universal adversarial perturbation):*** *$\boldsymbol{\delta}$ extracts the intrinsic vulnerability in the foundation model, which can mislead the $f_\phi$ successfully on most natural inputs $\boldsymbol{x}$, which is:*

$$\max_{\boldsymbol{\delta} \in \mathcal{B}_\epsilon} \mathop{\mathbb{E}}_{\boldsymbol{x} \sim D} \left[ \mathbb{I} \left\{ \mathcal{L} \left( f_\phi \left( \boldsymbol{x} \right), f_\phi \left( \boldsymbol{x} + \boldsymbol{\delta} \right) \right) > \lambda \right\} \right],  \tag{8}$$

   *where $\lambda$ is a pre-defined threshold for one successful attack, and $\mathbb{I} \{\cdot\}$ is the indicator function that returns 1 if the inside condition is satisfied, else 0.*

2. ***Fast adaptability (meta-initialization):*** *for any downstream task $\tau$ with the corresponding private downstream dataset $D_\tau$ and model $f_{\phi_\tau}$, the attackers can maximize the loss $\mathcal{L}$ on downstream model $f_{\phi_\tau}$ by updating the initialization $\boldsymbol{\delta}$ via the surrogate model $f_\phi$ in $t$ steps, which is:*

$$\max_{\boldsymbol{\delta} \in \mathcal{B}_\epsilon} \mathop{\mathbb{E}}_{\boldsymbol{x}_\tau \sim D_\tau} \left[ \mathcal{L}_\tau \left( f_{\phi_\tau} \left( \boldsymbol{x}_\tau \right), f_{\phi_\tau} \left( \boldsymbol{x}_\tau + U^t \left( \boldsymbol{\delta} \right) \right) \right) \right],  \tag{9}$$

   *$U^t$ is the operation to update $\boldsymbol{\delta}$ for $t$ steps based on input $\boldsymbol{x}_\tau$, which is defined as:*

$$U^t \left( \boldsymbol{\delta} \right) = clip_{B_\epsilon} \left[ \boldsymbol{\delta} + \sum_{j=1}^{t} \Delta \boldsymbol{\delta}_j \right],  \tag{10}$$

   *where $\Delta \boldsymbol{\delta}_{j+1} = \alpha_\tau \cdot sign \left( \nabla \mathcal{L} \left( f_\phi \left( \boldsymbol{x}_\tau \right), f_\phi \left( \boldsymbol{x}_\tau + U^j \left( \boldsymbol{\delta} \right) \right) \right) \right)$ if we attack the surrogate model $f_\phi$ and update the adversarial perturbation based on the first-order gradient.*

Generally, Equation 8 aims to extract the intrinsic vulnerability inherent in the model, which remains effective towards the input variation. Equation 9 guarantees that utilizing the perturbation $\boldsymbol{\delta}$ as the initialization can rapidly threaten strong adversarial attacks for any downstream model. However, in Equation 9, $D_\tau$ and $f_{\phi_\tau}$ are unknown if the attacker is precluded from the downstream dataset. An approximated solution to that involves using the dataset $D$ that covers the distribution of most natural inputs and a general model $f_\phi$ that can approximately represent the expectation of $f_{\phi_\tau}$, which is:

$$\max_{\boldsymbol{\delta} \in \mathcal{B}_\epsilon} \mathop{\mathbb{E}}_{\boldsymbol{x} \sim D} \left[ \mathcal{L} \left( f_\phi \left( \boldsymbol{x} \right), f_\phi \left( \boldsymbol{x} + U^t \left( \boldsymbol{\delta} \right) \right) \right) \right].  \tag{11}$$

To optimize the above two objectives simultaneously, our learning aims to move towards the direction that maximizes the inner product of the gradients computed on both objectives. We utilize a first-order meta-learning algorithm called Reptile [41], which defines the noise $\boldsymbol{\delta}$ update in each round as:

$$\boldsymbol{\delta} = \boldsymbol{\delta} + \eta \cdot \frac{1}{n} \sum_{i=1}^{n} \left( \tilde{\boldsymbol{\delta}}_{\mu_i} - \boldsymbol{\delta} \right),  \tag{12}$$

where $\eta$ is the update step size, and $\tilde{\boldsymbol{\delta}}_{\mu_i} = U^t_{\mu_i} \left( \boldsymbol{\delta} \right)$ is the updated perturbation on objective $\mu_i$ after optimizing $t$ iterations. Here we set $n = 2$, corresponding to the two objectives in Equation 8 and 11. For $\mu_1$ that aims to optimize Equation 11, we set $t = 5$ and $U^t_{\mu_1} \left( \boldsymbol{\delta} \right)$ the same as $U^t$ defined in Equation 10. For $\mu_2$ that aims to optimize Equation 8, we set $U^t_{\mu_2} \left( \boldsymbol{\delta} \right)$ as:

$$U^t_{\mu_2} \left( \boldsymbol{\delta} \right) \leftarrow \arg \min_{\boldsymbol{\delta} + \Delta \boldsymbol{\delta}} \| \Delta \boldsymbol{\delta} \|_\infty, \ \text{s.t.} \mathcal{L} \left( f_\phi \left( \boldsymbol{x} \right), f_\phi \left( x + \boldsymbol{\delta} + \Delta \boldsymbol{\delta} \right) \right) > \lambda, \ \Delta \boldsymbol{\delta} = \sum_{j=1}^{t} \Delta \boldsymbol{\delta}_j.  \tag{13}$$

Equation 13 aims to find a minimal update $\Delta \boldsymbol{\delta}$ nearby $\boldsymbol{\delta}$ to mislead the model $f_\phi$. This can be achieved by using enough iterations $t$ and a small but gradually increased norm-ball boundary $\epsilon$. While finding an effective UMI requires a substantial number of inputs and iterations, this process can be conducted fully offline, thus not hindering real-time adversarial attacks.

### 4.3 Enhance the transferability via gradient robust loss

Besides the utilization of intrinsic weakness inherent in the foundation model to enhance the adversarial attack $\mathcal{AT}^*$, another method involves generating the adversarial perturbation that sustains robustness against the deviation arising from updates through a surrogate model that exhibits significant gradient disparity compared to the fine-tuned downstream model.

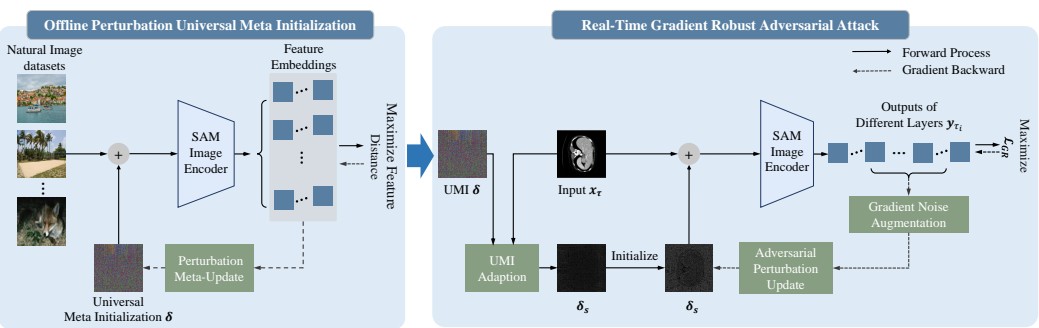

Figure 2: The data flow of our UMI-GRAT, consisting of an offline learning process of UMI and a real-time gradient robust adversarial attack.

Let us first assume that the surrogate model $f_\phi$ consists of $m$ sequentially connected modules, denoted as $\{f_{\phi^1}^1, \ldots, f_{\phi^m}^m\}$. The outputs of those modules are denoted as $\{y^1, \ldots, y^m\}$, with $y^i = f_\phi^i(y^{i-1})$. For the victim model $f_{\phi_\tau}$ with updated parameter $\Delta\phi_\tau$, the modules are denoted as $\{f_{\phi^1+\Delta\phi_\tau^1}^1, \ldots, f_{\phi^m+\Delta\phi_\tau^m}^m\}$. The output $y_\tau$ of each module with the input $x_\tau$ is denoted as:

$$y_\tau^i = f_{\phi^i+\Delta\phi_\tau^i}^i\left(y_\tau^{i-1}\right) = f_{\phi^i}^i\left(y_\tau^{i-1}\right) + h_{\Delta\phi_\tau^i}^i\left(y_\tau^{i-1}\right), \tag{14}$$

where $y_\tau^0 = x_\tau$ and $h_{\Delta\phi_\tau^i}^i$ is a hypothetical function that characterizes the update brought by $\Delta\phi_\tau^i$.

**Proposition 1 (Deviation in updating adversarial perturbation).** *Let $f_{\phi_\tau}$ be the victim model fine-tuned on any unknown task $\tau$, the deviation in the direction of updating the adversarial perturbation by maximizing a predefined loss $\mathcal{L}$ in the surrogate model $f_\phi$ can be formulated as:*

$$\Delta\delta_\tau - \Delta\delta_s = \nabla\mathcal{L}(y_\tau^m) \cdot \left(\prod_{i=1}^m \left(\nabla f_{\phi^i}^i\left(y_\tau^{i-1}\right) + \nabla h_{\Delta\phi_\tau^i}^i\left(y_\tau^{i-1}\right)\right) - \prod_{i=1}^m \nabla f_{\phi^i}^i\left(y_\tau^{i-1}\right)\right). \tag{15}$$

In Equation 15, $\Delta\delta_\tau \leftarrow \nabla\mathcal{L}(y_\tau^m) \cdot \prod_{i=1}^m \left(\nabla f_{\phi^i}^i\left(y_\tau^{i-1}\right) + \nabla h_{\Delta\phi_\tau^i}^i\left(y_\tau^{i-1}\right)\right)$ is the update of adversarial perturbation if white-box attack the victim model and $\Delta\delta_s \leftarrow \nabla\mathcal{L}(y_\tau^m) \cdot \prod_{i=1}^m \nabla f_{\phi^i}^i\left(y_\tau^{i-1}\right)$ is the update of adversarial perturbation by maximizing the pre-defined $\mathcal{L}$ on feature embeddings of the surrogate model. Proposition 1 establishes by simultaneously considering the white-box scenarios for both surrogate and victim models to derive the gradient using the chain rule. It claims that $h_{\Delta\phi_\tau^i}^i$ leads to a great deviation in updating the adversarial perturbation $\delta_s$ towards the optimal solution in attacking victim model if directly maximizing the feature embedding distance of the surrogate model, thus degrading the effectiveness of the generated AEs.

**Mitigate the deviation caused by gradient disparity.** To enhance effectiveness of the generated adversarial perturbation $\delta_s$ under the hypothetical update $\nabla h_{\Delta\phi_\tau}$, we propose a gradient robust loss $\mathcal{L}_{GR}$, that aims to mitigate the deviation in Equation 15 by gradient-based noise augmentation. Denote $\mathcal{N}(\varepsilon; \mu, \sigma^2 \boldsymbol{I})$ as the isotropic Gaussian noise with mean $\mu$ and variance $\sigma^2$, which has the same dimension as $\nabla h_{\Delta\phi_\tau}$. The robust update of adversarial perturbation $\Delta\delta_s^*$ on the surrogate model based on the noised augmented gradient is:

$$\Delta\delta_s^* \leftarrow \nabla\mathcal{L}(y_\tau^m) \cdot \prod_{i=1}^m \left(\nabla f_{\phi^i}^i\left(y_\tau^{i-1}\right) + \varepsilon_i \cdot \nabla f_{\phi^i}^i\left(y_\tau^{i-1}\right)\right). \tag{16}$$

By ignoring higher-order uncertain terms in Equation 16, we can simplify it as:

$$\Delta\delta_s^* \leftarrow \nabla\mathcal{L}(y_\tau^m) \cdot \left(\prod_{i=1}^m \nabla f_{\phi^i}^i\left(y_\tau^{i-1}\right) + \sum_{i=1}^m \varepsilon_i \cdot \prod_{j=1}^{i-1} \nabla f_{\phi^j}^j\left(y_\tau^{j-1}\right)\right). \tag{17}$$

Following the adversarial perturbation update guidance in Equation 17, the corresponding gradient robust loss $\mathcal{L}_{GR}$ is defined as :

$$\mathcal{L}_{GR} = \left\| f_{\phi^m}^m\left(y_\tau^{m_{adv}}\right) - f_{\phi^m}^m\left(y_\tau^m\right) + \frac{1}{m-1}\sum_{i=1}^{m-1} \varepsilon_i \cdot \left(f_{\phi^i}^i\left(y_\tau^{i_{adv}}\right) - f_{\phi^i}^i\left(y_\tau^i\right)\right)\right\|_p, \tag{18}$$

**Algorithm 1** Generating adversarial examples by UMI-GRAT

1: **Input:** open-sourced foundation model $f_\phi$, natural image dataset $D$, task-specific image $\boldsymbol{x}_\tau$, number of meta-iterations $T_m$, universal step size $\eta$, attack iterations $T_a$, attack step size $\alpha$.

2:   # Off-line learning a UMI
3:   **function** UNI_META_INI($f_\phi, D, T_m, \eta$)
4:     **Initialize** $\boldsymbol{\delta}, \boldsymbol{\delta}_{\mu_1}, \boldsymbol{\delta}_{\mu_2}$ with **0**
5:     **for** t $\leftarrow$ 1 to $T_m$ **do**
6:       assign $\boldsymbol{\delta}_{\mu_1}, \boldsymbol{\delta}_{\mu_2} = \boldsymbol{\delta}$
7:       **for** each $\boldsymbol{x}$ in $D$ **do**
8:         update $\tilde{\boldsymbol{\delta}}_{\mu_1}$ by $U_{\mu_1}^t(\tilde{\boldsymbol{\delta}}_{\mu_1})$ in Equation 10
9:         update $\tilde{\boldsymbol{\delta}}_{\mu_2}$ by $U_{\mu_2}^t(\tilde{\boldsymbol{\delta}}_{\mu_2})$ in Equation 13
10:    update $\boldsymbol{\delta} = \boldsymbol{\delta} + \eta \cdot \frac{1}{2}\sum_{i=1}^{2}(\tilde{\boldsymbol{\delta}}_{\mu_i} - \boldsymbol{\delta})$
11:   **return** $\boldsymbol{\delta}$

12:   # Real-time attack by UMI-GRAT
13:   **function** GR_ATTACK($f_\phi, \boldsymbol{x}_\tau, T, \eta, \boldsymbol{\delta}$)
14:     # adapt the universal perturbation $\boldsymbol{\delta}$
15:     $\boldsymbol{\delta}_{adp} = \alpha_{adp} \cdot sign\left(\nabla\mathcal{L}_{adp}\left(f_\phi\left(\boldsymbol{x}_\tau\right), \tilde{\boldsymbol{y}}\right)\right)$
16:     **Initialize** $\boldsymbol{\delta}_s^* = clip_{B_\varepsilon}\left[\boldsymbol{\delta} + \boldsymbol{\delta}_{adp}\right]$
17:     # threaten GRAT
18:     **for** t $\leftarrow$ 1 to $T_a$ **do**
19:       $\boldsymbol{g}_s \leftarrow \nabla\mathcal{L}_{GR}\left(f_\phi\left(\boldsymbol{x}_\tau + \boldsymbol{\delta}_s^*\right), f_\phi\left(\boldsymbol{x}_\tau\right)\right)$
20:       $\boldsymbol{\delta}_s^* = clip_{B_\varepsilon}\left[\boldsymbol{\delta}_s^* + \alpha \cdot sign(\boldsymbol{g}_s)\right]$
21:     $\boldsymbol{x}_\tau^{adv} = \boldsymbol{x}_\tau + \boldsymbol{\delta}_s^*$
22:     **return** $\boldsymbol{x}_\tau^{adv}$

where $\boldsymbol{y}^{i_{adv}}$ is the extracted adversarial feature by layer $i$ and $\|\cdot\|_p$ is a predefined norm-based measure that is decided by the $\mathcal{L}$ (*e.g.*, $p = 1$ for L1 loss).

**Discussion with the intermediate-level attacks**. The intermediate-level attacks (ILAs) [34, 22, 32] also aim to maximize the dissimilarity of feature embeddings between the clean and adversarial inputs. However, the main concern in ILAs is how to find a directional vector $\boldsymbol{v}$ to guide the update direction of $f_\phi(\boldsymbol{x}) - f_\phi(\boldsymbol{x}^{adv})$, thus assuring that this feature-wised dissimilarity can maximally mislead the final prediction. Different from that, our $\mathcal{L}_{GR}$ considers the problem one step further: given an optimal directional vector $\boldsymbol{v}$, how to generate

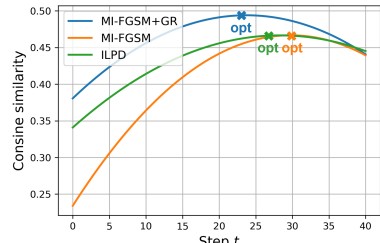

Figure 3: The cosine similarity of white-box generated perturbations on surrogate and victim models.

the adversarial perturbation that is roust towards the potential gradient variation in the victim model, thus maximally misleading the victim model along the direction of $\boldsymbol{v}$. Our core idea is hence in parallel with ILAs and can be well combined with them to enhance the attack's effectiveness. We experimentally analyze and visualize the cosine similarity of the generated perturbations on CT-scan images by white-box attacking open-sourced SAM and medical SAM using MI-FGSM [13], ILPD [34], and our gradient robust attack in Figure 3, illustrating that the proposed $\mathcal{L}_{GR}$ effectively reduce the deviation caused by gradient variation and achieves a much transferability.

# 5   Implementation of the proposed MUI-GRAT

The detailed implementation of our proposed MUI-GRAT is illustrated in Algorithm 1 and Figure 2. Our UMI-GRAT method consists of two stages. The initial stage involves the offline learning of a universal meta-initialization (UMI), which aims to find the intrinsic vulnerability inherent in the foundation model. In the subsequent stage, we utilize the learned UMI as the prior knowledge to enhance the gradient-variation robust adversarial attack.

We utilize the image encoder from Vit-H-based SAM as the general foundation model $f_\phi$ for generating the UMI. The natural image dataset $D$ consists of a total of 20,000 images, with 10,000 from ImageNet and 10,000 from the SA-1B dataset. We set the meta iterations $T_m$ as 7 and the universal step size $\eta$ as 1. The function Uni_Meta_Ini returns the learned UMI $\boldsymbol{\delta}$ that can be used to enhance the generation of subsequent input-specific adversarial perturbation.

In GR_Attack, we first adapt the calculated UMI $\boldsymbol{\delta}$ with the task-specific image $\boldsymbol{x}_\tau$ by one-step update using FGSM [18] with the step size $\alpha_{adp} = 4$. Assume that $\tilde{\boldsymbol{y}}$ represents the mean of the feature embedding of natural images calculated by $f_\phi$, our $\mathcal{L}_{adp}$ is defined as:

$$\mathcal{L}_{adp} = \|mean(f_\phi(\boldsymbol{x}_\tau)) - \tilde{\boldsymbol{y}}\|_p. \tag{19}$$

Equation 19 aims to minimize the domain difference between $v\boldsymbol{x}_\tau$ and the natural images by a specific generated perturbation. The UMI $\boldsymbol{\delta}$ is then added by $\boldsymbol{\delta}_{adp}$ and utilized to initialize $\boldsymbol{\delta}_s^*$. We set $T_a$ to 10, and update the adversarial perturbation $\boldsymbol{\delta}_s^*$ by maximizing the gradient robust loss $\mathcal{L}_{GR}$.

Table 1: Comparison results of transfer-based adversarial attacks on different models. The surrogate model is the open-sourced SAM.

| Model | Medical SAM [61] | | Shadow-SAM [11] | Camouflaged-SAM [11] | | | | | |
|---|---|---|---|---|---|---|---|---|---|
| Dataset | CT-Scans | | ISTD | COD10K | | CAMO | | CHAME | |
| Metrics | mDSC↓ | mHD↑ | BER↑ | $S_\alpha\downarrow$ | MAE↑ | $S_\alpha\downarrow$ | MAE↑ | $S_\alpha\downarrow$ | MAE↑ |
| Without attacks | 81.88 | 20.64 | 1.43 | 0.883 | 0.025 | 0.847 | 0.070 | 0.896 | 0.033 |
| MI-FGSM [13] | 40.83 | 64.42 | 4.31 | 0.372 | 0.214 | 0.331 | 0.286 | 0.352 | 0.250 |
| DMI-FGSM [56] | 34.51 | 74.20 | 4.39 | 0.435 | 0.134 | 0.395 | 0.210 | 0.416 | 0.164 |
| PGN [17] | 43.15 | 58.03 | 5.16 | 0.368 | 0.230 | 0.336 | **0.318** | 0.340 | 0.275 |
| BSR [50] | 25.7 | 94.48 | 5.20 | 0.414 | 0.146 | 0.372 | 0.226 | 0.402 | 0.178 |
| ILPD [34] | 33.65 | 65.98 | 4.40 | 0.366 | 0.245 | **0.310** | 0.316 | 0.327 | 0.287 |
| MUI-GRAT | **5.22** | **111.87** | **12.46** | **0.360** | **0.248** | 0.329 | 0.308 | **0.332** | **0.293** |
| MUI-GRAT+DMI-FGSM | 5.28 | 114.68 | 5.48 | 0.409 | 0.198 | 0.417 | 0.267 | 0.406 | 0.228 |
| MUI-GRAT+PGN | 9.62 | 115.87 | **33.98** | 0.358 | 0.262 | 0.353 | 0.306 | 0.332 | 0.296 |
| MUI-GRAT+BSR | 3.61 | 105.31 | 7.00 | 0.385 | 0.219 | 0.398 | 0.277 | 0.387 | 0.245 |
| MUI-GRAT+ILPD | **3.52** | **121.89** | 15.56 | **0.349** | **0.263** | **0.321** | **0.311** | **0.315** | **0.317** |

# 6 Experimental Results

## 6.1 Experiment setup

**Evaluation details:** we conduct experiments on SAMs' downstream models including, medical image segmentation SAM [61], shadow segmentation SAM [11], and camouflaged object segmentation SAM [11]. The datasets include: the synapse multi-organ segmentation dataset [29] that contains 3779 abdominal CT scans with 13 types of organs annotated, the ISTD dataset [49] that contains 1870 image triplets of shadow images, the COD10K dataset [14] that contains 5066 camouflaged object images, the CHAMELEON dataset that contains 76 camouflaged images, and the CAMO dataset [30], that contains 1500 camouflaged object images. We report the mean dice similarity score (mDSC) and mean Hausdorff distance (mHD) for evaluating medical segmentation, mean absolute error (MAE) and structural similarity ($S_\alpha$) for camouflaged object segmentation, and the bit error rate (BER) for shadow segmentation. In medical SAM, the image encoder is based on SAM-Vit-B and fine-tuned with LoRA [21]. In shadow segmentation and camouflaged object segmentation SAM, the image encoders are based on SAM-Vit-H and fine-tuned with the adapter [20]. The decoders in those models are all fully retrained.

**Compared methods:** We mainly compare and evaluate our method with current transfer-based adversarial attacks including gradient-based attacks called MI-FGSM [13] and PGN [17], input-augmentation based attacks called DMI-FGSM [56] and BSR [50], and intermediate-level feature based attack called ILPD [34].

**Implementation details:** we use the MI-FGSM [13] as our basic attack method. For all methods reported, we set the attack update iterations $T_a$ as 10, with the $l_\infty$ bound $\epsilon = 10$ and the step size $\alpha = 2$. For our UMI, we set the meta iterations $T_m = 7$, universal step size $\eta = 1$. For PGN and BSR, we set the number of examples as 8 for efficiency.

## 6.2 Main results

We report our main results in Table 1. The first row of the data presents the model performance with clean inputs. The second part of the data shows the model performance under different adversarial attacks, where the data with the strongest attack is bold. The results demonstrate that the adversarial examples generated by our proposed MUI-GRAT are more effective and generalizable than others, consistently posing significant adversarial threats across various downstream models. In medical segmentation and shadow segmentation tasks that share a great difference with the natural segmentation tasks, our proposed MUI-GRAT greatly surpasses others (*e.g.,* MUI-GRAT reduces the mDSC from 81.88 to 5.22 while the previous best is 25.73.). This demonstrates the exceptional effectiveness of our proposed MUI-GRAT when the attacker lacks information about the victim model, thereby generating AEs using a surrogate model distinct from the victim model. In the camouflaged object segmentation task where the data closely resembles natural images, all methods exhibit strong transferability. Our MUI-GRAT achieves the best performance on the COD10K and CHAME datasets and performs comparably to the SOTA method on the CAMO dataset.

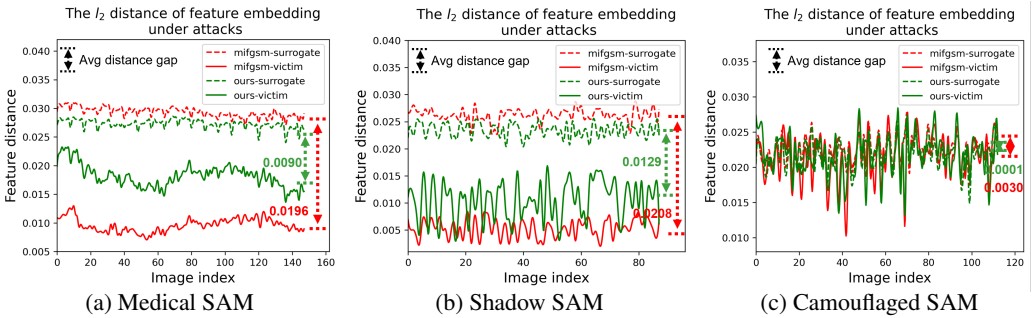

Figure 4: The $l_2$ distance of feature embedding from clean inputs and adversarial examples. The small distance gap between the surrogate and victim models indicates better transferability.

In the last part of Table 1, we analyze the performance of our proposed MUI-GRAT when combined with other SOTA transfer-based attacks. Notably, all methods achieve an overall performance gain after combining with ours. Though a slight performance drop occurs when attacking the CAMO dataset, combining the MUI-GRAT brings great performance gain on attacking other tasks. This demonstrates the versatility of our MUI-GRAT, which can be seamlessly applied in a plug-and-play manner to bolster existing transfer-based attacks. We report the experiment results for attacking open-sourced SAMs in Appendix A.2 and analyze the real-time attack efficiency for each method in Appendix A.3.

## 6.3 Analysis of the transferability

The transferability property across diverse models ensures the effectiveness of the adversarial example to mislead an unknown victim model. As discussed in Section 4.1, an intuitive objective for attacking SAM's downstream models is to maximize the dissimilarity of feature embedding extracted from the clean input $x$ and adversarial input $x^{adv}$. Based on this, to numerically evaluate the improvement of transferability brought by MUI-GRAT, we present the $l_2$ distance of clean and adversarial feature embeddings attacked by MI-FGSM and ours and then analyze the distance gap between the surrogate and victim models. We propose that a viable transferable attack methodology should induce a substantial feature distance $\Delta f$ in the victim model, while simultaneously ensuring minimal performance degradation $\epsilon$ during the transition from the surrogate to the victim model.

We show this comparison result in Figure 4, where we randomly pick a subset of inputs and show the distance of feature embedding for each clean-adversarial input pair. The average distance gap between the surrogate and victim models indicates the overall transferability of the attack method. In the medical and shadow segmentation SAM, where the data and task are distinct from the original SAM, we find a great performance drop for MI-FGSM when transferred from the surrogate model to the victim model. Though the adversarial examples generated by MI-FGSM induce a large feature distance in the surrogate model, their effect on the victim model is relatively minor. Conversely, the adversaries generated by MUI-GRAT maintain much better transferability, suffering from a small performance drop when transferred from the surrogate model to the victim model. In the camouflage object segmentation task, where the data are natural images and the segmentation objective is similar to the original SAM, both attack algorithms show good transferability (nearly no performance drop when transferred from the surrogate model to the victim models). Our MUI-GRAT shows better transferability with nearly no performance drop.

## 6.4 Ablation study

In this section, we explore the contribution of the proposed MUI and GRAT by integrating them with MI-FGSM and PGN attacks. The ablation results are shown in Table 2. In scenarios where the task and dataset distributions of surrogate and victim models differ markedly (e.g., medical image segmentation and natural image segmentation), we observe that the GR loss significantly enhances effectiveness. Meanwhile, across all scenarios, the proposed MUI consistently contributes to enhancing the adversarial attacks. Particularly in camouflaged object segmentation when the surrogate and victim models exhibit close similarities, the MUI yields substantial benefits.

Table 2: The performance of methods combined by MUI and GR.

| Basic strategy | MUI | GR | Medical SAM [61] | | Shadow-SAM [11] | Camouflaged-SAM [11] | |
|---|---|---|---|---|---|---|---|
| | | | mDSC↓ | mHD↑ | BER↑ | $S_\alpha$↓ | MAE↑ |
| MI-FGSM [13] | ✗ | ✗ | 40.83 | 64.42 | 4.31 | 37.17 | 21.41 |
| | ✓ | ✗ | 37.54 | 70.13 | 4.72 | **36.11** | **24.74** |
| | ✗ | ✓ | **6.34** | **100.52** | **8.07** | 36.90 | 21.78 |
| PGN [17] | ✗ | ✗ | 43.15 | 58.03 | 5.16 | 36.84 | 23.04 |
| | ✓ | ✗ | 41.13 | 66.11 | 6.46 | **35.23** | **27.68** |
| | ✗ | ✓ | **15.51** | **93.98** | **10.36** | 36.25 | 24.10 |

This observation aligns with our analysis in Section 4. By assuming a hypothetical update $h_{\Delta\phi_\tau}$ in the victim model, the proposed gradient robust loss greatly enhances the effectiveness of the generated AEs towards the gradient variation, thus benefiting more for medical and shadow segmentation tasks. Moreover, the MUI aims to find the intrinsic vulnerability inherent in the basic foundation model through a broad general dataset, which is then provided as the prior knowledge for generating a more effective adversary. Therefore, in scenarios where the victim model inherits substantial information from the surrogate model, this prior knowledge becomes increasingly reliable and effective.

## 7 Conclusion

The security of utilizing large foundation models is a critical issue for deploying them in real-world applications. This paper, for the first time, considers a more challenging and practical attack scenario where the attacker executes a potent adversarial attack on SAM-based downstream models without prior knowledge of the task and data distribution. To achieve that, we propose a universal meta-initialization (UMI) algorithm to uncover the intrinsic vulnerabilities inherent in the foundation model. Moreover, by theoretically formulating the adversarial update deviation during the attacking process between the open-sourced SAM and its fine-tuned downstream models, we propose a gradient robust loss that simulates the corresponding uncertainty with gradient-based noise augmentation and analytically demonstrates that the proposed method effectively enhances the transferability. Extensive experiments validate the effectiveness of the proposed UMI-GRAT toward SAM and its downstream tasks, highlighting the vulnerabilities and potential security risks of the direct utilization and fine-tuning of open-sourced large foundation models.

## Acknowledgment

This research is supported by the National Research Foundation, Singapore, and Infocomm Media Development Authority under its Trust Tech Funding Initiative, and by a donation from the Ng Teng Fong Charitable Foundation. Any opinions, findings and conclusions or recommendations expressed in this material are those of the author(s) and do not reflect the views of National Research Foundation, Singapore, and Infocomm Media Development Authority. This research is partly supported by the Program of Beijing Municipal Science and Technology Commission Foundation (No.Z241100003524010), and is partly supported by Guangdong Basic and Applied Basic Research Foundation (2024A1515010454). The research was carried out at the ROSE Lab, Nanyang Technological University, Singapore.

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

# A  Appendix

## A.1  Randomness test

Table A.3: Experimental randomness of transfer-based adversarial attacks on SAMs (subset).

| Model | Medical SAM | | Shadow-SAM | Camouflaged-SAM | | | | | |
|---|---|---|---|---|---|---|---|---|---|
| Dataset | CT-Scans | | ISTD | COD10K | | CAMO | | CHAME | |
| Metrics | mDSC↓ | mHD↑ | BER↑ | $S_\alpha$↓ | MAE↑ | $S_\alpha$↓ | MAE↑ | $S_\alpha$↓ | MAE↑ |
| MI-FGSM | 31.35(±1.11) | 73.26(±7.67) | 3.05(±0.26) | 0.37(±4e−3) | 0.23(±4e−3) | 0.33(±7e−3) | 0.27(±6e−3) | 0.37(±1e−3) | 0.24(±4e−3) |
| DMI-FGSM | 27.19(±1.00) | 87.54(±8.085) | 2.53(±0.16) | 0.45(±5e−3) | 0.11(±3e−3) | 0.39(±4e−3) | 0.20(±3e−3) | 0.43(±3e−3) | 0.14(±2e−3) |
| PGN | 40.64(±1.23) | 40.76(±8.95) | 3.05(±0.11) | 0.36(±5e−3) | 0.25(±8e−3) | 0.32(±6e−3) | **0.33**(±5e−3) | 0.36(±3e−3) | 0.23(±7e−3) |
| BSR | 28.62(±1.41) | 93.41(±5.82) | 2.58(±0.15) | 0.45(±6e−3) | 0.11(±3e−3) | 0.40(±3e−3) | 0.20(±2e−3) | 0.43(±4e−3) | 0.14(±3e−3) |
| ILPD | 28.68(±2.18) | 63.53(±9.22) | 2.74(±0.15) | 0.35(±2e−3) | 0.25(±7e−3) | **0.33**(±2e−3) | 0.28(±3e−3) | 0.37(±2e−3) | 0.24(±4e−3) |
| MUI-GRAT | **2.01**(±0.28) | **148.18**(±10.20) | **10.10**(±0.28) | **0.29**(±0.01) | **0.40**(±5e−3) | 0.31(±5e−3) | 0.33(±4e−3) | **0.33**(±5e−3) | **0.28**(±8e−3) |
| MUI-GRAT+DMI | 2.23(±0.37) | 123.76(±24.88) | 4.52(±0.20) | 0.40(±4e−3) | 0.20(±4e−3) | 0.37(±4e−3) | 0.27(±2e−3) | 0.38(±5e−3) | 0.23(±8e−3) |
| MUI-GRAT+PGN | 15.64(±0.63) | 112.38(±10.55) | **16.41**(±0.79) | 0.33(±7e−3) | 0.33(±0.01) | 0.31(±0.01) | 0.36(±0.01) | 0.34(±4e−3) | 0.27(±8e−3) |
| MUI-GRAT+BSR | 6.25(±0.58) | 82.56(±10.75) | 3.87(±0.17) | 0.41(±4e−3) | 0.21(±0.01) | 0.38(±6e−3) | 0.27(±6e−3) | 0.40(±0.01) | 0.22(±0.01) |
| MUI-GRAT+ILPD | **0.95**(±0.27) | **165.30**(±16.28) | 10.67(±0.52) | **0.28**(±7e−3) | **0.40**(±0.01) | **0.30**(±4e−3) | 0.33(±7e−3) | **0.34**(±4e−3) | **0.28**(±0.01) |

We evaluated 10 attack methods presented in our paper over 5 random seed runs on the subset of SAM's downstream tasks and reported the mean performance with its standard deviation. We use the same experimental setting provided in Section 6.1. The results indicate that the randomness is small and similar among all attacking methods. The uncertainty level of UMI-GRAT in mean Hausdorff Distance (mHD) is marginally higher compared to the other methods. This can account for the higher mHD value achieved by the UMI-GRAT.

## A.2  Attacking open-sourced SAMs

Table A.4: The performance of open-sourced SAMs under attacks.

| Surrogate model | Attack | SAM-Vit-B | | SAM-Vit-L | | SAM-Vit-H | |
|---|---|---|---|---|---|---|---|
| | | mAP↓ | mIOU↓ | mAP↓ | mIOU↓ | mAP↓ | mIOU↓ |
| **SAM-Vit-B** | MI-FGSM | 1.24 | 5.23 | 14.17 | 24.12 | 24.20 | 34.96 |
| | MUI-GRAT | **0.65** | **2.21** | **3.67** | **9.27** | **8.43** | **15.41** |
| **SAM-Vit-L** | MI-FGSM | **14.18** | **21.11** | **1.11** | 4.03 | **16.42** | **25.10** |
| | MUI-GRAT | 16.41 | 23.05 | 1.39 | **2.40** | 18.12 | 28.67 |
| **SAM-Vit-H** | MI-FGSM | 21.74 | 29.35 | 15.58 | 24.71 | 1.46 | 6.51 |
| | MUI-GRAT | **18.50** | **25.26** | **10.00** | **18.12** | **0.58** | **2.25** |

We report the mean average precision (mAP) and mean intersection-over-union (mIOU) metrics to evaluate the performance of attacking open-sourced SAMs. We compare the performance of MI-FGSM and ours MUI-GRAT. The implementation of these attacks adheres to the details described in Section 6.1. We randomly select 500 images from the SA-1B dataset and evaluate the performance of SAMs under the 'AutomaticMaskGenerator' mode. We white-box attacks SAM-Vit-B, SAM-Vit-L, and SAM-Vit-H models. Meanwhile, we also discuss the black-box transferability of generated AEs across different models. The results are shown in Table A.4.

Our findings reveal that both MI-FGSM and MUI-GRAT achieve comparably strong attack performance in the white-box scenario. In the black-box scenario, where AEs generated on one surrogate model are transferred to a different victim model, our results indicate that the surrogate model is critical for the transferability of generated AEs. The AEs with the strongest transferability are generated by our MUI-GRAT on attacking SAM-Vit-B, which exhibits a great performance gain compared with others. However, when employing SAM-ViT-L as the surrogate model, there is a slight reduction in the transferability of AEs produced by our MUI-GRAT. Conversely, when the surrogate model is SAM-Vit-H, the transferability of AEs generated by our MUI-GRAT surpasses MI-FGSM by a large margin.

## A.3 Analysis of the attack efficiency

We analyze the real-time attack efficiency of the methods mentioned above. We report the average time required for generating one AE when the input resolution is $512 \times 512$ on the SAM-Vit-B model using one RTX 4090 GPU. The results are shown in Table A.5.

We find that our proposed MUI-GRAT achieves second-high efficiency when compared with others. The ILPD requires extra attack iterations (e.g., 10-step MI-FGSM) to find a directional guide vector $v$. The input

Table A.5: Analysis of the attack efficiency

| Method | Num. of samples needed per iteration | Avg. time (s) |
|---|---|---|
| MI-FGSM | 1 | **0.32** |
| DMI-FGSM | 2 | 0.69 |
| PGN | 8 | 3.96 |
| BSR | 8 | 2.91 |
| ILPD | 1 | 0.66 |
| MUI-GRAT | 1 | 0.36 |

augmentation-based attack methods, such as BSR and DMI-FGSM, require multiple samples in each iteration to do the augmentation and the PGN also requires multiple samples to obtain a stable gradient direction. However, our method utilizes an offline generated MUI and only conducts one-time gradient augmentation in each iteration, thus achieving a much higher efficiency than others.

## A.4 Hardware Setup

We run our experiments for attacking the medical segmentation model using one RTX 4090 GPU with 24 GB memory. We run the rest of the experiments using one RTX A6000 GPU with 48 GB memory.

## A.5 Visualization of the adversarial examples and the prediction

We visualize adversarial examples generated by MUI-GRAT utilizing solely the open-sourced SAM and the corresponding adversarial predictions in Figure A.5 and Figure A.6. The images are randomly selected. This visualization provides a more straightforward demonstration of the impact of the adversarial attack threatened by MUI-GRAT, showing how it significantly compromises the reliability of the large foundation model's predictions with just a single independent input image.

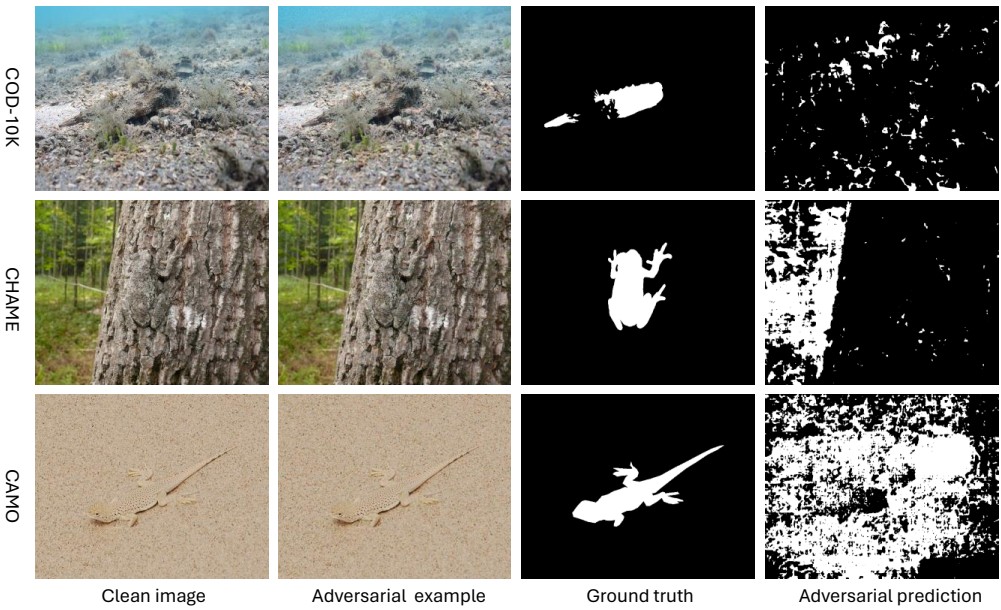

Figure A.5: The visualized adversarial attack results in camouflaged object segmentation task.

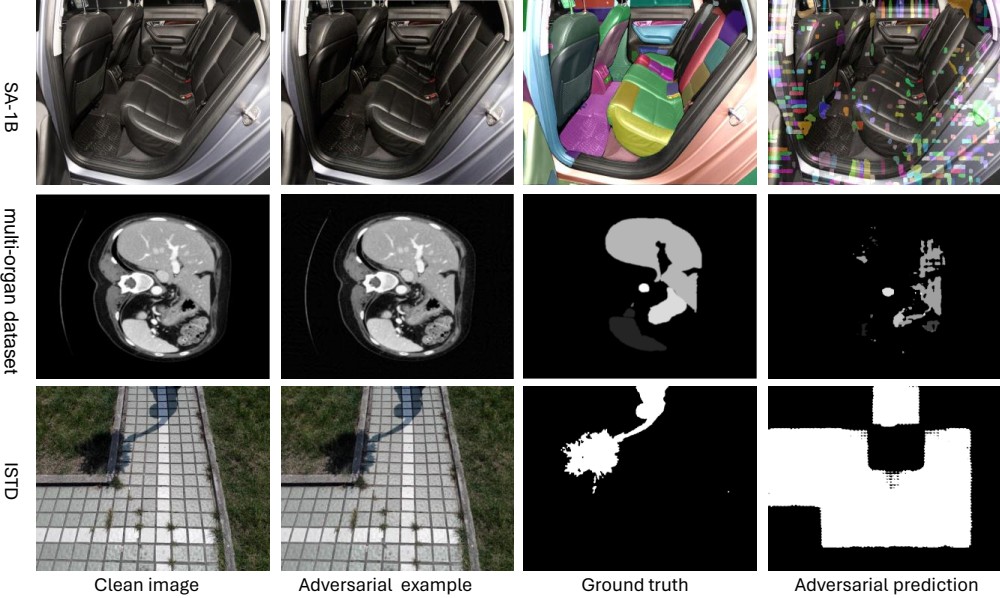

Figure A.6: The visualized adversarial attack results in natural, medical, and shadow image segmentation tasks.

### A.6 Impact Statement

The increasing reliance on large foundation models in various real-world applications amplifies the critical importance of ensuring their secure utilization. This paper mainly discusses the potential risks of adversarial threats associated with the direct utilization and fine-tuning of the open-sourced model even on private and encrypted datasets. To accomplish that, we begin an investigation into a more practical while challenging adversarial attack problem: attacking various SAM's downstream models by solely utilizing the information from the open-sourced SAM. We then provide the theoretical insights and build the experimental setting and benchmark, aiming to serve as a preliminary exploration for future research in this area. Experimentally, we validate the vulnerability of SAM and its downstream models under the proposed MUI-GRAT, indicating the security risk inherent in the direct utilization and fine-tuning of open-sourced large foundation models, thus highlighting the urgent need for robust defense mechanisms to protect these models from adversarial threats.

### A.7 Limitations

The limitations of our paper are:

- The proposed UMI-GRAT is not contingent upon a prior regarding the model's architecture, suggesting its potential applicability across various model paradigms. However, the experiments only tested MUI-GRAT on the prevalent SAMs and their downstream models. The capability of UMI-GRAT to pose a threat to other large foundation models remains a topic for further exploration.

- While this paper highlights the risk of direct utilization of SAM and fine-tuning it on the downstream task, this paper does not provide and validate an effective solution for this secure concern.

