# OpenReview forum: "Transferable Adversarial Attacks on SAM and Its Downstream Models"
_NeurIPS.cc/2024/Conference — NeurIPS 2024 poster_

### Official Review · Reviewer_aXPE · 2024-07-09

**Soundness:** 3
**Presentation:** 3
**Contribution:** 3
**Rating:** 8
**Confidence:** 4

**Summary:**

This work discusses an interesting security issue of deploying a model fine-tuned on a large foundational model in private downstream applications. It proposes a universal meta-initialized and gradient robust adversarial attack (UMI-GRAT) to break the powerful SAM and its various downstream models, without requiring prior knowledge of the specific downstream task and data distribution. The author explores the challenges associated with threating transfer-based adversarial attack without the task-related prior knowledge and provides the theoretical insights on the deviation in updating the adversarial perturbation when using the open-sourced model as the surrogate model. An extensive evaluation of UMI-GRAT's performance, transferability, and efficiency was conducted across five datasets and three different downstream tasks (medical image segmentation, shadow segmentation and camouflaged object segmentation), demonstrating the high effectiveness of the UMI-GRAT approach.

**Strengths:**

1. This work discusses a critical adversarial issue of deploying large foundation model in real-world applications and for the first time considers a more challenging and practical scenario where the adversarial attacker breaks SAM and its downstream models in the absence of prior knowledge on the task and data distribution.
2. This work provides the in-depth analysis on the challenge of threating the transferable adversarial attack via open-sourced SAM and proposes the corresponding theoretical insights and solution.
3. The work establishes a detailed experimental framework and the proposed UMI-GRAT shows superior performance on misleading various SAMs’ downstream models compared with previous methods, which serve as a preliminary exploration for future research.

**Weaknesses:**

1.	It’s recommended to give more comprehensive analysis of the UMI noise, including the size of the natural image dataset and the effect of various hyperparameters.
2.	There are more metrics such as $E_\phi$, $F_\beta^\omega$ in the camouflaged object detection task. It would be beneficial if the author could provide further data pertaining to these evaluation metrics to enrich the analysis.

**Questions:**

1.	In Figure 3, the peak cosine similarity of the generated adversarial perturbation is observed between the 20th and 30th iteration. So will increasing the iterative step of generating adversarial perturbation enhance the transferability?
2.	Model ensemble is an effective method to enhance the adversarial attacks’ transferability. Will the ensemble of different SAMs benefit the UMI-GRAT?

**Limitations:**

The experimental results are all based on SAMs and their downstream models. It will provide more valuable insights if expanding the scope of analysis to assess whether this adversarial threat also applies to other large foundation models.

---

> ### Author Rebuttal · Authors · 2024-08-06
>
> Dear Reviewer aXPE,
>
> Thank you so much for taking the time to read this paper and giving constructive feedback. Please find our response below.
>
> > Give a more comprehensive analysis of the UMI noise.
>
> Following your valuable suggestion, we conducted an evaluation to analyze the impact of the natural dataset size $N$ and the total meta iteration times $T_m$ on the UMI using the COD10K dataset.  We presented the results in the following table.
>
> |  Hyperparameters  | $S_\alpha\downarrow$ | MAE$\uparrow$ |
> | :---------------: | :------------------: | :-----------: |
> | $T_m=7,N=20,000$  |        36.11         |   **24.74**   |
> |   $T_m=7,N=200$   |        37.54         |     20.83     |
> |  $T_m=7,N=5,000$  |        36.81         |     23.23     |
> | $T_m=3,N=20,000$  |        36.43         |     24.30     |
> | $T_m=5,N=20,000$  |      **36.01**       |     24.58     |
> | $T_m=11,N=20,000$ |        36.59         |     24.59     |
>
> The results indicate that the size of the dataset $N$ has a significant impact on the UMI. As a smaller dataset can introduce substantial bias to the generated UMI, this thereby leads to the degraded performance of the generated UMI. The effect of  $T_m$ becomes minor when larger than 5.
>
> > Provide further data pertaining to $E_\Phi $ and $F_\omega ^\beta $.
>
> We presented the performance of $E_\Phi $ and $F_\omega ^\beta $ on the COD10K dataset under five different attacks in the following table:
>
> | Attack strategy | $E_\Phi\downarrow$ | $F_\omega ^\beta\downarrow$ |
> | :-------------: | :----------------: | :-------------------------: |
> | Without attack  |       0.918        |            0.801            |
> |     MI-FGSM     |       0.569        |            0.019            |
> |       PGN       |       0.537        |            0.021            |
> |       BSR       |       0.513        |            0.058            |
> |      ILPD       |       0.502        |          **0.017**          |
> |    UMI-GRAT     |     **0.478**      |            0.019            |
>
> The results show that our method achieves the best attack performance in terms of the E-measure ($E_\Phi$) and the second-best performance on the weight F-measure $F_\omega ^\beta $.  We will add those metrics in the table of our revised paper for better clarification.
>
> > Will increasing the iterative step of generating adversarial perturbation enhance the transferability?
>
> We conducted comparative experiments under 5 different attacks with iteration times $T_a=20$ and the results are reported in **Table R2** of our submitted PDF file.
>
> The results indicate that increasing the attack iterations **greatly enhances** the adversarial examples on attacking camouflaged SAM, where the **domain gap from the natural images is minor**. However, For those tasks characterized by a substantial domain gap, increasing the iteration cannot bring performance gain.
>
> > Will the ensemble of different SAMs benefit the UMI-GRAT?
>
> Following your valuable suggestion, we explored enhancing adversarial attacks by the ensemble under two types of scenarios:
>
> 1. The transferability from pre-trained SAM to Medical-SAM.
> 2. The transferability from the pre-trained SAM ViT-B to SAM ViT-H.
>
> We adhered to the experimental settings outlined in Section 6.1 and employed an ensemble of **SAM ViT-B and SAM ViT-L**. We reported the mDSC for Medical SAM and mIOU for the original SAM. The results are presented in the following tables:
>
> |                     | Medical-SAM(mDSC$\downarrow$ ) | SAM ViT-H(mIOU$\downarrow$ ) |
> | :-----------------: | :----------------------------: | :--------------------------: |
> |      UMI-GRAT       |              5.22              |            15.41             |
> | UMI-GRAT + Ensemble |             13.54              |            12.09             |
>
> The results indicate that the ensemble yields performance gains in general transfer tasks while degrading the performance on the pre-trained model to fine-tuned model transfer task. As the fine-tuned model solely inherits information from its pre-trained one, incorporating uncorrelated gradient information brings unnecessary deviation, thus degrades the performance.
>
> > Expanding the scope of analysis to assess whether this adversarial threat also applies to other large foundation models.
>
>  Following your good suggestion, we evaluated the effectiveness of our proposed UMI-GRAT on two new scenarios:
>
> 1. **The transferability of MUI-GRAT to other pre-trained methods.** We attacked the **MAE ViT-S and ViT-B that are fine-tuned on Chexpert [2]** using solely **the pre-trained MAE ViT-S** and reported the results in **Table. R3** of our submitted PDF file.
> 2. **The effectiveness of MUI-GRAT on the general transfer attack setting.** We attacked the **MAE ViT-B and DenseNet-121** **fine-tuned** **on Chexpert [2]** using **MAE ViT-S fine-tuned** on the same dataset and reported the results in **Table. R4**.
>
> We utilized models provided by [1] and followed the same experimental setting. We evaluate 8 attack methods on the Chexpert [2] dataset, where the model needs to diagnose five predefined diseases in the chest X-ray images.  We reported the mean Area Under the Curve (mAUC) to evaluate the performance.
>
> The results in **Table. R3** shows that the proposed **MUI-GART maintains the best method** on attacking both the fine-tuned **MAE-ViT-S and MAE-ViT-B** models. In **Table. R4**, we evaluated the transferability between different ViTs and transferability from the ViT to the CNN.  The experimental results indicate that our MUI-GRAT maintains effectiveness on general transfer tasks, demonstrating its good generalizability.
>
> ------
>
> We do appreciate your constructive feedback. We will add those experiments and analyses mentioned above in the appendix of our revised version.
>
> [1]  Delving into masked autoencoders for multi-label thorax disease classification. In WACV 2023.
>
> [2]  Chexpert: A large chest radiograph dataset with uncertainty labels and expert comparison. In AAAI 2019.

---

> > ### Comment · Reviewer_aXPE · 2024-08-07
> > **Response to the Authors**
> >
> > After carefully reviewing the comments from the other reviewers and the author's rebuttal, all of my concerns have been adequately addressed. Therefore, I decide to raise my score.

---

> > > ### Author Response · Authors · 2024-08-07
> > > **Response to reviewer aXPE**
> > >
> > > Dear Reviewer aXPE,
> > >
> > > Thank you so much for your positive and constructive feedback, which is very helpful and makes our paper stronger!
> > >
> > > We are glad that our responses address your concern. We are always available and eager to address any additional questions you might have during our discussion.
> > >
> > > Best regards,
> > >
> > > The Authors

---

### Official Review · Reviewer_iuQV · 2024-07-13

**Soundness:** 3
**Presentation:** 2
**Contribution:** 2
**Rating:** 4
**Confidence:** 3

**Summary:**

In this paper, the authors present a new approach for adversarial attacks on Segment Anything Model (SAM)-based downstream models, addressing the challenge of attacking without prior knowledge of the downstream task or data distribution. Their key contribution is a universal meta initialization-based algorithm that exposes inherent vulnerabilities in the foundation model. The authors also introduce a gradient robust loss, which simulates uncertainty through gradient-based noise augmentation. This loss is derived from a theoretical formulation of adversarial update deviation between the open-sourced SAM and its fine-tuned downstream models. The authors provide an analytical demonstration of how their proposed method enhances attack transferability. The effectiveness of their approach is thoroughly validated through comprehensive experiments.

**Strengths:**

Originality: This is the first work to explore the feasibility of adversarially attacking various downstream models fine-tuned from the Segment Anything Model (SAM). The introduction of a universal meta initialization-based algorithm to uncover intrinsic vulnerabilities in foundation models is both effective and efficient. Additionally, the formulation of adversarial update deviation and the proposal of a gradient robust loss that simulates uncertainty with gradient-based noise augmentation further enhance the transferability of adversarial examples.

Quality and Clarity: The writing is generally clear but has room for improvement. The methodology and results are well-structured, though some technical sections could benefit from additional clarification.

Significance: This work is highly significant given the increasing prevalence of foundation models like SAM. The proposed methods for enhancing attack transferability have important implications for AI system security and could influence future directions in both offensive and defensive strategies in adversarial machine learning for SAM.

**Weaknesses:**

1 - My major concern is related to the novelty of the proposed approach. Although I agree that this is the first work in the context of SAMs, the main components, such as downstream agnostic adversarial examples and meta learning-based fast initialization, have already been proposed in the literature.

2 - The authors, in line 45, briefly highlight downstream agnostic examples in just one line. They should clarify in the related work section how their work is different from references 55 and 56 of the main paper, beyond just applying it to SAM. Similarly, another related work that the authors missed is [1] (given below), in which the generated adversarial examples are agnostic to downstream tasks.

3 - Similarly, the authors did not mention any work related to meta-learning-based adversarial examples in the paper. There are multiple works that use meta-learning to craft universal adversarial examples, such as [1, 2] below. The authors use these meta-learning-based methods for initialization of adversarial examples, but this has already been explored in [3] below. The authors should mention these meta-learning-based approaches in their paper and discuss how their method is different from these approaches, beyond just the application to SAMs.

4 - It is not clear to me when the authors claim in line 8 that they are attacking "without accessing the downstream task." What is the task here? Is it not the segmentation task? In [1], their task-agnostic adversarial examples are effective against classification, detection, and segmentation. Since the downstream task here is segmentation-based, is it not obvious what the task is? Please clarify this.

5 - The authors should include some specific aspects of SAM to make their attack more unique. Currently, they are utilizing the SAM image encoder, which, in my opinion, is not much different from the previous works listed below.

6 - For experiments, why have the authors compared their method with intermediate-level feature-based approaches? They should also compare it with different downstream agnostic adversarial approaches as listed below.

7 - In Equation 8, how did the authors choose the threshold lambda?

[1] A Self-supervised Approach for Adversarial Robustness (CVPR-2020)

[2] Learning to Generate Image Source-Agnostic Universal Adversarial Perturbations (IJCAI22)

[3] Meta Adversarial Perturbations (AAI2022-Workshop)

[4] Adversarial Initialization with Universal Adversarial Perturbation: A New Approach to Fast Adversarial Training

**Questions:**

Please see the weakness section. While the paper presents an approach to attacking SAM-based downstream models, it largely combines existing methods rather than introducing new techniques. The current strategy, though effective, does not fully exploit SAM's unique architecture.

**Limitations:**

Yes.

---

> ### Author Rebuttal · Authors · 2024-08-06
>
> Dear reviewer iuQV
>
> Thank you so much for taking the time to read this paper and giving constructive feedback. Please find our response below.
>
> > Q1, Q2, and Q3: The novelty of the proposed method. The authors should discuss how their method is different from meta-learning-based approaches [2,3,4]. Clarify how their work is different from references 55 and 56. Another work[1] is missing.
>
> Although utilizing the UAP as the initial point is discussed in [2,3,4] and downstream-agnostic attacks are discussed in [1, 55, 56], the core methodology design in our UMI is significantly different from previous work. Moreover, the proposition of GRAT in mitigating the gradient misalignment is unique, novel, and effective. The main differences are:
>
> 1. **Compared with meta-learning approaches:** [3] utilizes the UAP to generate AEs with a **one-step update**, and [4] uses UAP for **adversarial training**. [2] enhances the existing UAP under the **few-shot learning** scenarios:  learning a UAP with a few examples to attack the same victim model. Different from previous work, we first utilize the UMI to extract the intrinsic vulnerability **inherent in the pre-trained foundation model** and utilize it as the prior knowledge to enhance the **attack on fine-tuned downstream models**.
>
> 2. **Compared with model-agnostic approaches:** [1] first proposed a representation-based adversarial attack to enhance the downstream-agnostic adversarial training. [55] proposed the first framework for generating downstream-agnostic UAP on self-learning models and [56] extends this attack to multimodality. However, those methods do not consider the utilization of **intrinsic vulnerability in the pre-trained model and the gradient misalignment brought by fine-tuning**, thus do not work well when the downstream dataset **exhibits a distinctive domain gap** from the pre-trained dataset (e.g. from a natural dataset to the medical dataset).
>
> The novelty of our work is two-fold:
>
> 1.  **Exploitation of intrinsic vulnerability via UMI**: we first utilize the UMI to extract the **intrinsic vulnerability inherent in the pre-trained foundation model** and utilize it to enhance the attack on fine-tuned downstream models.
>
> 2.  **Rectification of gradient misalignment via GRAT**: Inspired by our **Proposition 1, which formulates the deviation** occurred on attacking the unknown fine-tuned model, we propose the **GRAT to effectively mitigate this misalignment**. The experimental results in Figure 4 and Table 2 demonstrate that **when the fine-tuned model has a pronounced gradient update, the proposed GRAT greatly rectifies the deviation and brings a great performance gain**.
>
>
> We will discuss and cite all the mentioned work above in our revised paper.
>
> > Q2 and Q6: Why compare with intermediate-level feature-based approaches? They should also compare it with different downstream agnostic approaches.
>
> **The Reason for comparison with ILPD is**: as shown in Equation (7) of the submitted paper, our attack aims to **maximize the feature distance**. We thus discuss and compare our method with the SOTA feature-based attack.
>
> **Comparison with [1]**: according to Section 3.1 and Equation (4) in [1], the attack of [1] directly perturbs the input to **maximize feature distortion via MI-FGSM**. According to Equation (7) of the submitted paper, all methods compared in Table 1 aim to **maximize the feature distortion of SAM's image encoder**. This means **the compared MI-FGSM in Table 1 is totally the same as the attack algorithm proposed in [1]**. We will add more explanations in our revised version to dispel this confusion.
>
> **Comparison with [55, 56]**: as [55, 56] focus on **universal adversarial perturbation**, directly comparing them with **the input-specific attacks**  is somewhat unfair to them. We evaluated the Adv-PER [55] in attacking **normally trained and adversarial-trained Medical SAMs** and reported the experimental results in **Table R5** of our submitted PDF file, indicating that the Adv-PER does not work well when the downstream dataset exhibits a significant domain gap from the pre-trained dataset.
>
> > Q4. Clarify the claim "without accessing the downstream task".
>
> Our attack perturbs the feature encoder of SAM, which, by fine-tuning the decoder, can zero-shot and few-shot transfer to various **downstream tasks, such as edge prediction**. **Without accessing the task, the distortion in the feature encoder** will adversely affect the performance across different types of decoders. To substantiate this and demonstrate the generation, we conducted a new experiment on attacking the MAE model fine-tuned on the Chexpert [5], a **task for diagnosing chest X-ray diseases**. We reported the results in **Table R3 and R4** of our submitted PDF, demonstrating the effectiveness and generalization of our method across different downstream tasks.
>
> > Q5. The authors should include some specific aspects of SAM to make their attack more unique.
>
> As stated in lines 548-549 in our submitted paper, our proposed UMI-GRAT is not contingent upon a prior on model’s architecture, suggesting its potential applicability across various model paradigms. The experiments on attacking the fine-tuned MAE ViTs and CNNs in Tables R3 and R4 demonstrate its good generalization. Following your suggestion, we will clarify this in our revised version.
>
> > Q7. How to choose the threshold lambda in Equation 9?
>
> Thanks for your good suggestion, we will add the discussion below in our revised paper:
>
> "We initialize the $\lambda$ with a value of 0.05, which is easily satisfied by most inputs, at the first epoch and increase it by a factor of 2 if at least 50% of the inputs meet this threshold."
>
>
> ---
>
> We greatly appreciate your constructive feedback and will add all the experiments with analyses and cite all mentioned papers in our revised version.
>
> [5]  Chexpert: A large chest radiograph dataset with uncertainty labels and expert comparison. In AAAI 2019.

---

> > ### Comment · Reviewer_iuQV · 2024-08-13
> >
> > The authors have addressed most of my concerns. While this is the first work in the context of SAM, components like meta-learning-based adversarial examples already exist in the literature and should be properly credited. Additionally, related work [1] should be properly cited. I have raised the score and hope the authors will open-source their code for reproducibility.

---

> > > ### Author Response · Authors · 2024-08-13
> > > **Response to Reviewer iuQV**
> > >
> > > Thank you for your positive feedback. We are glad that our responses address your concerns. In our revised version, we will properly cite all the mentioned papers [1,2,3,4]. All codes will be open-sourced for reproducibility.
> > >
> > > We are always available and eager to address any further questions you may have during our discussion.

---

### Official Review · Reviewer_Q28B · 2024-07-13

**Soundness:** 3
**Presentation:** 3
**Contribution:** 3
**Rating:** 7
**Confidence:** 4

**Summary:**

This paper proposes an adversarial attack against fine-tuned derivatives to a publicly available foundation model, such as the Segment Anything Model (SAM). In the proposed threat model, attackers can potentially manipulate these downstream models even without knowing the specific task or data they are used for. Under this threat model, proposes a new attack method called UMI-GRAT (Universal Meta-initialized and Gradient Robust Adversarial Attack). Through a bi-level optimization procedure, this method leverages the information from the open-source SAM to create adversarial examples that can fool mislead the original SAM and its fine-tuned versions. Finally, this paper demonstrates the effectiveness of the proposed UMI-GRAT attack against SAM through extensive experiments.

**Strengths:**

1. The paper is motivated by real-world safety concerns for fine-tuning a public foundation model on private domain-specific datasets.
2. The figures and tables are well-polished and generally reflect the overall message of the paper.
3. The proposed UMI-GRAT attack method is unique and backed by theoretical analysis.

**Weaknesses:**

1. The effectiveness of the proposed attack is only demonstrated by attacking the SAM model. However, more experiment settings (e.g. against pretrained MAE models) are warranted to demonstrate the generalizability of the proposed attack.

**Questions:**

1. How does the domain gap between the natural image dataset used to obtain the universal adversarial trigger and the downstream dataset influence the effectiveness of the attack?
2. How effective is the proposed method against adaptive defense? For example, if the downstream victim model has gone through adversarial training, how effective would the adversarial trigger obtained on the unguarded pretrained SAM be against the guarded victim model?

**Limitations:**

The authors acknowledge the limitations of this work in the appendix. They candidly acknowledge the limitations in evaluations as the proposed attack is only evaluated against SAM. I appreciate the authors openly acknowledging this limitation.

---

> ### Author Rebuttal · Authors · 2024-08-06
>
> Dear Reviewer Q28B,
>
> Thank you so much for taking the time to read this paper and giving constructive feedback. Please find our response below.
>
> > 1. More experiment settings (e.g. against pretrained MAE models) are warranted to demonstrate the generalizability.
>
>  Following your good suggestion, we evaluated the effectiveness of our proposed UMI-GRAT on two new scenarios:
>
> 1. **The transferability of MUI-GRAT to other pre-trained methods.** We attacked the **MAE ViT-S and ViT-B that are fine-tuned on Chexpert [2]** using solely **the pre-trained MAE ViT-S** and reported the results in **Table. R3** of our submitted PDF file.
> 2. **The effectiveness of MUI-GRAT on the general transfer attack setting.** We attacked the **MAE ViT-B and DenseNet-121** **fine-tuned** **on Chexpert [2]** using **MAE ViT-S fine-tuned** on the same dataset and reported the results in **Table. R4**.
>
> We utilized models provided by [1] and followed the same experimental setting. In both scenarios, we evaluate 8 attack methods on the Chexpert [2] dataset, where the model needs to diagnose five predefined diseases in the chest X-ray images.  Following [1], we report the mean Area Under the Curve (mAUC) to evaluate the performance. Each experiment is run for 5 times.
>
> The results in **Table. R3** shows that the proposed **MUI-GART maintains the best method** on attacking both the **MAE-ViT-S and MAE-ViT-B** models fine-tuned on the downstream data. Moreover, the combination of the MUI-GRAT with the second-best method BSR [3] brings a further enhancement to the performance.
>
> In **Table. R4**, we evaluated the transferability between different ViTs and transferability from the ViT to the CNN.  The experimental results indicate that our MUI-GRAT maintains effectiveness on general transfer tasks (MUI-GRAT achieves **the second-best** attack performance compared to other SOTA methods), demonstrating its good generalizability.
>
> > 2. How does the domain gap between the natural image dataset used to obtain the universal adversarial trigger and the downstream dataset influence the effectiveness of the attack?
>
> The efficacy of the universal adversarial trigger increases as the domain gap between the downstream dataset and the natural image dataset narrows. As mentioned in lines "334-340" of the submitted paper, the UMI extracts the intrinsic vulnerability inherent in the foundation model, thus being more effective when the victim model inherits significant information from the pre-trained model. The following table presents the performance of the SAM before and after being fine-tuned on downstream tasks:
>
> |                                  | Medical SAM (mDSC) | Camouflaged SAM (MAE) |
> | -------------------------------- | :----------------: | :-------------------: |
> | Performance pre/post-fine-tuning |     1.39/81.88     |      0.050/0.025      |
> | Attack performance gain by UMI   |        3.29        |       **2.98**        |
> | Attack performance gain by GRAT  |     **34.49**      |         0.37          |
>
> The results show that the fine-tuning brings a significant performance gain for Medical SAM but a minor performance gain for Camouflaged SAM, indicating that the domain gap between the medical and natural datasets is huge and the model is accompanied by a pronounced gradient update. The domain gap between the camouflaged and natural datasets is small so that the gradient modification following fine-tuning is minor. The performance gain shown in the rest data demonstrates that the UMI derives a greater advantage than the GRAT when the domain gap is small and GRAT displays a contrary trend, which aligns with our analysis in Section 4.
>
> > 3. How effective is the proposed method against adaptive defense?
>
> Following your constructive feedback, we conduct comparative experiments on attacking the **adversarial trained Medical SAM**. As our attacks aim to maximize the feature distance, we thereby consider two types of adversarial training (AT) mechanisms:
>
> 1. **Feature-wise AT**: the defender is **aware of feature-wise attacks**, thus assuming an attacker whose objective is to **maximize the distance of feature embedding** during AT, and hence the defender minimizes the adversarial loss.
> 2. **Output-wise AT**: the defender is **unaware of feature-wise attacks**, thus assuming an attacker whose objective is to **maximize the final segmentation loss** during AT, and hence the defender minimizes the adversarial segmentation loss.
>
> We optimize the Medical SAM by incorporating the adversarial loss into the training loss with a weight hyperparameter $\tau$. We evaluate $\tau=0.1\,0.5$ and use the MI-FGSM with iteration $T_a =1, 5$ with bound $\epsilon=10$ as the attacking strategy.
>
> We ran 9 different attacks on 6 different AT models on CT-Scan and presented the results in **Table R.5**. The results demonstrate that:
>
> 1. The proposed UMI-GRAT remains the most effective method even if the model has a certain robustness via AT, demonstrating the effectiveness of our proposed UMI-GRAT.
> 2. Feature-wise AT surpasses output-wise adversarial training for those feature maximizing attacks in most scenarios.
> 3. The robustness of AT varies drastically with different training hypermeters. Enlarging the weight $\tau$ benefits the robustness. An intriguing finding is that using a stronger gradient attack (e.g . MI-FGSM $T_a =1\,to \, 5$) during AT may damage the robustness towards **adversarial examples generated from the pre-trained model**.
>
> ---
>
> We greatly appreciate your constructive feedbacks. We will add those experiments and analyses mentioned above in the appendix of our revised version.
>
> [1] Delving into masked autoencoders for multi-label thorax disease classification. In WACV 2023.
>
> [2] Chexpert: A large chest radiograph dataset with uncertainty labels and expert comparison. In AAAI 2019.

---

> > ### Comment · Reviewer_Q28B · 2024-08-07
> > **Thanks for Your Response & Missing PDF**
> >
> > Thank you for your detailed responses and follow-up experiments! However, I couldn't find the PDF file you submitted. If you could provide me with a pointer to the revised PDF, I will make sure to go over it in the upcoming days. Thank you!

---

> ### Author Response · Authors · 2024-08-07
> **The missing PDF**
>
> Dear Reviewer,
>
> Thank you so much for your follow-up. The PDF file is attached to the overall Author Rebuttal. However, it seems that there is a bug in the OpenReview website that makes this rebuttal invisible to reviewers. We believe that NeurIPS will address this issue soon, and you can find the submitted PDF in the Author Rebuttal section.
>
> Best regards,
> The Authors

---

> > ### Comment · Reviewer_Q28B · 2024-08-07
> > **Comments Adequately Addressed**
> >
> > Thank you for the clarification! I can now see your follow-up PDF. I think my comments are adequately addressed, so I raise my score to seven.

---

> > > ### Author Response · Authors · 2024-08-07
> > > **Response to Reviewer Q28B**
> > >
> > > Dear Reviewer Q28B,
> > >
> > > Thank you so much for your positive and constructive feedback, which is very helpful and makes our paper stronger!
> > >
> > > We are glad that our responses address your concern. We are always available and eager to address any additional questions you might have during our discussion.
> > >
> > > Best regards,
> > >
> > > The Authors

---

### Official Review · Reviewer_Qxcd · 2024-07-13

**Soundness:** 3
**Presentation:** 3
**Contribution:** 2
**Rating:** 5
**Confidence:** 4

**Summary:**

This paper investigates the vulnerability of Segment Anything Model (SAM) and its downstream models to transferable adversarial attacks. The authors propose a novel attack method called Universal Meta-Initialized and Gradient Robust Adversarial attack (UMI-GRAT) that leverages the open-sourced SAM to generate adversarial examples effective against fine-tuned downstream models, even without access to the downstream task or dataset.

**Strengths:**

1. The paper tackles a practical and challenging problem of attacking downstream models fine-tuned from a publicly available foundation model without knowledge of the downstream task or data.
2. The proposed UMI-GRAT method is well-motivated and technically sound. The authors provide theoretical insights into the gradient deviation problem and propose a robust solution using gradient noise augmentation.
3. The paper presents extensive experiments demonstrating the effectiveness of UMI-GRAT in attacking SAM and its downstream models

**Weaknesses:**

See Questions.

**Questions:**

1. How does the performance of UMI-GRAT vary with different choices of hyperparameters, such as the perturbation bound ε and the number of iterations in UMI and LGR. Especially, line 278 mentions that the perturbation bound is 10, which is a bit too large.
2. According to the latest benchmark **[R1]**, baselines used in the paper are not SOTA methods. How does it compare with NCS **[R2]**, ANDA **[R3]**, DeCowA **[R4]** and L2T **[R5]**?
3. Is the proposed method a universal transfer attack method? Although this question is mentioned on line 548, can the performance of UMI-GRAT and SOTA be compared under a general transfer attack test setting?
4. UMI-GRAT is a gradient-based attack method. How does the proposed method perform when the model has a certain robustness (such as adversarial training)?

---
**[R1]** Devling into Adversarial Transferability on Image Classification: A Review, Benchmark and Evaluation.
**[R2]** Enhancing Adversarial Transferability Through Neighborhood Conditional Sampling.
**[R3]** Strong Transferable Adversarial Attacks via Ensembled Asymptotically Normal Distribution Learning. CVPR. 2024.
**[R4]** Boosting Adversarial Transferability across Model Genus by Deformation-Constrained Warping.
**[R5]** Learning to Transform Dynamically for Better Adversarial Transferability

**Limitations:**

Limitations are discussed in the Appendix.

---

> ### Author Rebuttal · Authors · 2024-08-06
>
> Dear Reviewer Qxcd,
>
> Thank you so much for taking the time to read this paper and giving constructive feedback. Please find our response below.
>
> > 1. How does the performance of UMI-GRAT vary with different hyperparameters, such as the bound $\epsilon$ and iterations?
>
> Following your good suggestion, we conducted experiments using two additional sets of hyperparameters: bound $\epsilon=4$ with iteration $T_a=10$ and bound $\epsilon=10$ with iteration $T_a=20$. We evaluated 5 different attacks and reported the experimental results in **Table R2** of our submitted PDF file. The results indicate that:
>
> 1. In datasets that exhibit a large domain gap from natural image datasets (e.g., medical and shadow datasets), the attack bound $\epsilon$ is critical for transferability. Reducing the bound leads to a substantial performance decline for all attack algorithms. For the camouflaged object segmentation that shares a small domain gap with the original SAM's task, reducing the norm bound only causes a marginal performance drop.
> 2. Increasing the attack iterations markedly enhances the AEs when the surrogate and victim domains are proximate. However, for those tasks characterized by a substantial domain gap, increasing the iteration has little help.
>
> > 2. Is the proposed method a universal transfer attack method? Can the UMI-GRAT and SOTA be compared under a general transfer attack test setting?
>
>  Following your good suggestion, we evaluated the effectiveness of our proposed UMI-GRAT on two new scenarios:
>
> 1. **The transferability of MUI-GRAT to other pre-trained methods.** We attacked the **MAE ViT-S and ViT-B that are fine-tuned on Chexpert [2]** using solely **the pre-trained MAE ViT-S** and reported the results in **Table. R3** of our submitted PDF file.
> 2. **The effectiveness of MUI-GRAT on the general transfer attack setting.** We attacked the **MAE ViT-B and MAE DenseNet-121** **fine-tuned** **on Chexpert [2]** using **MAE ViT-S fine-tuned** on the same dataset and reported the results in **Table. R4**.
>
> We utilized models provided by [1] and followed the same experimental setting. In both scenarios, we evaluate 8 attack methods on the Chexpert [2] dataset, where the model needs to diagnose five predefined diseases in the chest X-ray images.  We reported the mean Area Under the Curve (mAUC) to evaluate the performance.
>
> The results in **Table R3 and R4** demonstrate the great generalizability of our proposed method. **Table R3** shows that the proposed **MUI-GART maintains the best method** on attacking both the fine-tuned **MAE-ViT-S and MAE-ViT-B** models. In **Table. R4**, the evaluation of the transferability between different ViTs and transferability from the ViT to the CNN indicates that our MUI-GRAT maintains effectiveness (**the second-best method**) on general transfer tasks.
>
> > 3. How does it compare with NCS **[R2]**, ANDA **[R3]**, DeCowA **[R4]** and L2T **[R5]**
>
> Following your good suggestion, we evaluated two currently open-sourced methods, **ANDA[R3] and L2T[R5]**. We conducted experiments considering the above two scenarios along with attacking the **normally trained and adversarial trained Medical SAM**. We set the n_ens=5 for ANDA and num_ scale=3 for L2T and keep the rest hyperparameters the same. We reported the experimental results in **Table R3, R4, and R5** in our submitted PDF file.
>
> The results show that L2T performs well in both transfer tasks on MAE-based models while failing in attacking the SAM and adversarial-trained SAM. ANDA performs well on the general transfer tasks while failing on all pre-trained to fine-tuned transfer tasks. We hypothesize the reason for their failures as:
>
> 1.  Due to the great misalignment in the gradient discussed in Definition 1 and Proposition 1, the transfer between pre-trained and fine-tuned models is much harder than the general one.
> 2.  The feature-oriented attacking task may potentially undermine those output-oriented attack methods.
>
> > 4.  How does the proposed method perform when the model has a certain robustness (such as adversarial training)?
>
> Following your constructive feedback, we conducted comparative experiments on attacking the **adversarial trained Medical SAM**. As our attacks aim to maximize the feature distance, we thereby consider two types of adversarial training (AT) mechanisms:
>
> 1. **Feature-wise AT**: the defender is **aware of feature-wise attacks**, thus assuming an attacker to **maximize the distance of feature embedding** during AT, and the defender thereby minimizes the adversarial loss.
> 2. **Output-wise AT**: the defender is **unaware of feature-wise attacks**, thus assuming an attacker to **maximize the final segmentation loss** during AT, and the defender thereby minimizes the adversarial segmentation loss.
>
> We optimize the Medical SAM by incorporating the adversarial loss into the training loss with a weight factor $\tau$. We used $\tau=0.1\,0.5$ and took the MI-FGSM with iteration $T_a =1, 5$ and the same bound $\epsilon=10$ as the attacking strategy.
>
> We ran **9 different attacks on 6 different AT models** and presented the results in **Table R.5**, demonstrating that:
>
> 1. The proposed UMI-GRAT remains the most effective method even if the model has a certain robustness via AT, demonstrating the effectiveness of our proposed method.
> 2. Feature-wise AT surpasses output-wise AT towards those feature maximizing attacks in most scenarios.
> 3. Enlarging the weight $\tau$ benefits the robustness. An intriguing finding is that using a stronger gradient attack during AT (e.g . 5-steps MI-FGSM) may damage the robustness.
> ---
> We greatly appreciate your constructive feedback and will add all those experiments with analyses and cite all mentioned papers in our revised version.
>
> [1]  Delving into masked autoencoders for multi-label thorax disease classification. In WACV 2023.
>
> [2]  Chexpert: A large chest radiograph dataset with uncertainty labels and expert comparison. In AAAI 2019.

---

### Official Review · Reviewer_pDnZ · 2024-07-16

**Soundness:** 3
**Presentation:** 2
**Contribution:** 2
**Rating:** 5
**Confidence:** 3

**Summary:**

In this paper, the authors propose an adversarial attack method that can contaminate downstream tasks from the perspective of adversarial transferability. They address the problem that SAM models do not have similar optimisation routes after fine-tuning for different downstream tasks by designing universal meta initialization. In this paper, the authors address the problem that SAM models do not have similar optimisation routes after fine-tuning for different downstream tasks by designing UMI noise. The authors introduce the idea of meta-learning to allow their algorithm to quickly adapt to different situations, i.e., downstream tasks.

**Strengths:**

1. The theoretical part of this paper is detailed, the experiments are sufficient. The comparison with other methods shows the sophistication of their approach.



2. The attacks proposed in this paper are novel. It contributes to the topic of attacking downstream tasks of large models. A discussion on adversarial transferability is introduced under this topic.

**Weaknesses:**

1. The readability of the Methodology section of this article is somewhat poor. The authors define the problem to be solved through the form of propositions. Similarly, if the authors could summarise the formulas as Theorem and put the proof process (both formulas and reasoning) specifically in the supplementary material, it would make the article more coherent.



2. The randomness of the experimental results is unknown. I understand that due to the larger computational effort, it is not practical to report error lines on all major experiments. But it would be better for the authors to report a set of randomness on a smaller dataset and simpler settings, which will influence the reviewers' opinion of the results of this method.

**Questions:**

Two questions listed, see Weaknesses for details. Note that if the authors can demonstrate the randomness of their algorithms, that will help to get a higher rating.

**Limitations:**

The authors correctly list the implications of their work for the use of large models such as SAM in downstream tasks.

---

> ### Author Rebuttal · Authors · 2024-08-06
>
> Dear Reviewer pDnz,
>
> Thank you so much for taking the time to read this paper and giving constructive feedback. Please find our response below.
>
> > 1.The readability of the Methodology section of this article is somewhat poor. The authors define the problem to be solved through the form of propositions. Similarly, if the authors could summarise the formulas as Theorem and put the proof process (both formulas and reasoning) specifically in the supplementary material, it would make the article more coherent.
>
> Thanks for your constructive feedback. We will consolidate the Proposition and Equations (15) through (18) into a Theorem and put the remaining formulas and explanatory text in the appendix.
>
>
>
> > The randomness of the experimental results is unknown. It would be better to report a set of randomness experiments on a smaller dataset and simpler settings.
>
> Following your good suggestion, we evaluated **10** attack methods presented in our paper over **5** random seed runs on the subset of SAM's downstream tasks and reported the mean performance with its standard deviation. We use the same experimental setting provided in Section 6.1 of our submitted paper.
>
> The details of each subset are:
>
> 1. For the Medical SAM, we selected 'case0008' (comprising 148 images with a resolution of 512x512) from the validation set of the CT-Scan dataset.
> 2. For the Shadow SAM, we randomly selected 100 images with a resolution of 1024x1024 from the test set of the ISTD dataset.
> 3. For the Camouflaged SAM that is evaluated across three different datasets, we randomly selected 40 images with a resolution of 1024x1024 from each dataset (COD10k, CAMO, CHAME), with a total of 120 images.
>
> The results are shown in **Table R1** of our submitted **PDF file**,  which demonstrates that the randomness is small and similar among all attacking methods. The uncertainty level of UMI-GRAT in mean Hausdorff Distance (mHD) is marginally higher compared to the other methods. This can account for the higher mHD value achieved by the UMI-GRAT.
>
> We greatly appreciate your constructive feedback. We will conduct the randomness evaluation over the entire dataset and update the results in Table 1 and Table 2 of our paper in the revised version.

---

> > ### Comment · Reviewer_pDnZ · 2024-08-08
> > **Response to the Authors**
> >
> > It looks like the randomness of most of the data is controlled. The authors also promised to fix the mentioned errors in the next draft. Therefore I decided to raise my score.

---

> > > ### Author Response · Authors · 2024-08-08
> > > **Response to the Reviewer pDnZ**
> > >
> > > Dear Reviewer pDnZ,
> > >
> > > Thank you so much for your positive and constructive feedback, which is very helpful and makes our paper stronger!
> > >
> > > We are glad that our responses address your concern. We are always available and eager to address any additional questions you might have during our discussion.
> > >
> > > Best regards,
> > >
> > > The Authors

---

### Author Rebuttal · Authors · 2024-08-06

We express our sincere appreciation to all the reviewers for their elaborate and constructive feedback. We summarize our rebuttal as follows:

1. As suggested by reviewer **pDnz**, we conducted the **randomness experiment** and presented the experimental results in **Table R1** of the PDF document attached below.
2. As suggested by reviewer **Qxcd and aXPE**, we discussed the effect of different hyperparameters for different attack algorithms and presented the experimental results in **Table R2** of the PDF document attached below.
3. As suggested by reviewer **Qxcd**, we compared our method with **ANDA and L2T** and presented the experimental results in **Table R3, R4 and R5** of the PDF document attached below.
4. As suggested by reviewers **Qxcd, Q28B, and aXPE**, we evaluated our proposed method on **MAE pre-trained model and on a general transfer adversarial attack task**. The results are presented in **Tables R3 and R4** of the PDF document attached below.
5. As suggested by reviewers **Qxcd and Q28B**, we evaluated our proposed method on the **adversarial-trained Medical SAM** and reported the results in **Table R5** of the PDF document attached below.
6. As suggested by reviewer **iuQV**, we evaluated the mentioned **model-agnostic method** and presented the results in **Table R5** of the PDF document attached below.

---

### Decision · Program_Chairs · 2024-09-25

**Decision:**

Accept (poster)

**Comment:**

This paper explores the feasibility of adversarial attacks on downstream models fine-tuned from the SAM solely using information from the open-sourced SAM. They introduce a Universal Meta-Initialization and Gradient Robust Adversarial Attack (UMI-GRAT) to enhance the effectiveness of attacks on models fine-tuned on unknown datasets. By extracting intrinsic vulnerabilities from the foundation model and utilizing prior knowledge for adversarial perturbation generation, UMI-GRAT showcases improved transferability in attacking SAMs and their downstream models without accessing task-specific datasets.

On the positive side, the reviewers found the paper to be well-motivated and technically sound. The experimental results are sufficient and the proposed UMI-GRAT is well-supported by theoretical analysis. On the negative side, there are some concerns regarding the randomness, missing comparisons, and generalizability.

The authors have provided detailed responses. After fruitful discussions between reviewers and authors, these concerns were effectively addressed and most reviewers increased their scores. AC thus recommends acceptance.